# Intrinsic functional architecture of the non-human primate spinal cord derived from fMRI and electrophysiology

Tung-Lin Wu[1,2], Pai-Feng Yang[1,3], Feng Wang [1,3], Zhaoyue Shi[1,2], Arabinda Mishra[1,3], Ruiqi Wu[1], Li Min Chen [1,3] & John C. Gore[1,2,3,4]

Resting-state functional MRI (rsfMRI) has recently revealed correlated signals in the spinal cord horns of monkeys and humans. However, the interpretation of these rsfMRI correlations as indicators of functional connectivity in the spinal cord remains unclear. Here, we recorded stimulus-evoked and spontaneous spiking activity and local field potentials (LFPs) from monkey spinal cord in order to validate fMRI measures. We found that both BOLD and electrophysiological signals elicited by tactile stimulation co-localized to the ipsilateral dorsal horn. Temporal profiles of stimulus-evoked BOLD signals covaried with LFP and multiunit spiking in a similar way to those observed in the brain. Functional connectivity of dorsal horns exhibited a U-shaped profile along the dorsal-intermediate-ventral axis. Overall, these results suggest that there is an intrinsic functional architecture within the gray matter of a single spinal segment, and that rsfMRI signals at high field directly reflect this underlying spontaneous neuronal activity.

[1] Vanderbilt University Institute of Imaging Science, Nashville, TN 37232, USA. [2] Biomedical Engineering, Vanderbilt University, Nashville, TN 37232, USA. [3] Radiology and Radiological Sciences, Vanderbilt University Medical Center, Nashville, TN 37232, USA. [4] Department of Physics and Astronomy, Vanderbilt University, Nashville, TN 37232, USA. These authors jointly supervised this work: Li Min Chen, John C. Gore. Correspondence and requests for materials should be addressed to T.-L.W. (email: tung-lin.wu@vanderbilt.edu)

The identification of patterns of highly correlated, low-frequency fMRI signals in the brain in a resting state has provided a powerful approach to delineating functional architecture and neural circuits noninvasively[1–5]. Altered resting-state functional connectivity (rsFC) under different pathological conditions suggests that these correlations reflect intrinsic neural processes that are relevant for maintaining normal brain functions[6]. Over the past two decades, there have been thousands of reports of rsfMRI studies of the brain, but only recently has there been an emergence of similar studies in gray matter of the spinal cord[7–16]. This could be attributed to difficulties that arise in imaging the spinal cord—a small physical size, pronounced physiological noise, and increased effects from susceptibility gradients[17]. Nonetheless, recent technical advances have allowed robust detection of strong, rsFC between spinal horns in animal models as well as human subjects[7,8,10,13,15,16,18]. These observations have led several research groups to hypothesize that, like the brain, spinal cord gray matter exhibits its own intrinsic functional architecture that serves as a fundamental framework for executing and maintaining sensory, motor, and autonomic functions. However, given the complex and indirect coupling of blood-oxygen-level-dependent (BOLD) fMRI signals with electrical activity, validation of their relevance by comparisons to direct measurements of neural activity is critical to allow more precise interpretations of rsfMRI correlations within the spinal cord.

Previous studies have demonstrated a direct link between neural activity and BOLD signals in regions of the brain[19–24] in stimulation and task conditions. However, direct comparisons of spontaneous electrophysiological activity and rsfMRI signals remain relatively unexplored in the brain[20,21,25,26], and even less is known for the spinal cord. In the rat, previous studies have described stimulus- or movement-induced electrophysiological activities in the spinal cord[27,28]. In particular, Brieu et al. measured changes in blood volume and flow using intrinsic optical imaging following electrical stimulation, and subsequently compared optical responses with electrophysiological measurements[29]. While this study provides complementary insights to the work presented here, no previous study has directly compared and related rsfMRI signals to neural activity within the spinal gray matter in a resting state. The current study aims to address three key questions: (1) what are the intrinsic functional organization features of the spinal gray matter, (2) what is the relationship between changes in stimulus-evoked fMRI signals and the underlying neural electrical activity (local field potentials (LFPs) and multiunit activity (MUA)), and (3) do correlations in rsfMRI signal fluctuations (indicators of resting state FC) reflect the underlying synchronous variations in spontaneous neural electrophysiological activity within the spinal gray matter from the same regions.

In this study, we combined measurements of LFPs and MUA in non-human primates (NHPs) to validate findings from stimulus-driven and resting-state fMRI signal changes. Strong agreements were found between the two modalities for both the spatial patterns of metrics of neural activities and their inter-regional correlations: tactile stimulation evoked activations at ipsilateral horns in both fMRI and electrical recordings, while resting-state correlations between regions showed parallel relationships. This study identifies an intrinsic functional architecture relating dorsal–intermediate–ventral areas, and validates the interpretation of rsfMRI studies of the spinal cord. We propose that this will provide a foundation for using fMRI for assessing and monitoring alterations in spinal cord circuits that occur with injury and other pathologies.

## Results

**Comparable tactile responses between the two modalities.** To determine the relationship between spinal fMRI signals and neural electrophysiological activity during the processing of innocuous tactile inputs, we recorded and compared the temporal variations of BOLD, LFP, and MUA signals during the presentation of 8-Hz vibration stimulations of a single distal pad in a block design experiment. Figure 1e illustrates the quality of our sub-millimeter fMRI acquisitions, and Fig. 1g shows an example of the average BOLD activation map from one monkey obtained at 9.4 T. Figure 1f shows that 8-Hz innocuous tactile stimulation of a single digit elicited the strongest BOLD signal changes in the ipsilateral dorsal horn, with a peak value of 0.76 ± 0.06%. Figure 2 shows representative stimulus-driven LFP and multiunit spike activity recorded from the ipsilateral horn that receives inputs from the stimulated digits, which also demonstrate strong responses to stimulation. LFP voltage changes followed the stimulus time course (yellow shades in Fig. 2b, e) and frequency peaks are apparent at harmonics of 8 Hz in the LFP power spectra (Fig. 2c, f). This observation was consistent among all four monkeys that underwent electrophysiological testing. Group-normalized LFP power changes showed digit-selective responses to the tactile stimulus (compare the four different color curves in Fig. 2a, d). Similarly, in line with the LFP observation, spike rate histograms as well as peri-event raster plots of spike density in the ipsilateral horn also showed the largest responses to the single digit that was stimulated (Fig. 2h, k). Group-normalized spike rates obtained from four different recording sites (left D3, right D3, left D5, and right D5) are presented in Fig. 2g, j. Figure 3 presents group-averaged BOLD signal time courses and neural signal (LFP and MUA) responses to tactile stimulation overlaid on the same plot. Overall, significant and correlated changes in all three measures were observed in the ipsilateral dorsal horn of the homeotropically appropriate cervical segment, a finding that is congruent with fMRI results here and in previously reported studies[30,31].

**Local dorsal spatial resting-state correlation profiles.** Previous studies in NHP spinal cords revealed a strong functional connectivity between dorsal–dorsal and dorsal–ventral horns. Taking advantage of the high signal-to-noise ratio (SNR) and the resolution of high-field rsfMRI data, we plotted the local correlation profiles of the signals from a dorsal horn seed region along the dorsal–intermediate–ventral axis. Small regions of interest (ROIs) were drawn from the dorsal toward the ventral horn on one side of the spinal cord (Fig. 4a). In over half of the observations (58% of the runs), we found that correlations demonstrated a U-shaped pattern in which correlation strengths decreased monotonically from a high value to a low value at the level of the intermediate gray matter, and then increased as the ventral region was approached (Fig. 4b, c and Supplementary Figure 1). In the remaining runs, the correlations decreased continuously. This suggests the existence of three distinct subdivisions (dorsal, intermediate, and ventral), and allowed the intermediate GM seed ROIs to be more confidently defined for subsequent comparisons with electrophysiology (see details in the Methods section). This local correlation profile was supported at the group level after aligning the defined intermediate GM ROIs shown in Fig. 4c. The temporal SNR (tSNR) of the voxels examined did not show systematic changes as a function of distance from the seed (Fig. 4d), suggesting that the correlation profile was not driven by varying noise levels across three zones.

**Validation of rsfMRI connectivity of the spinal horns.** Inter-horn resting-state connectivity patterns observed in rsfMRI were

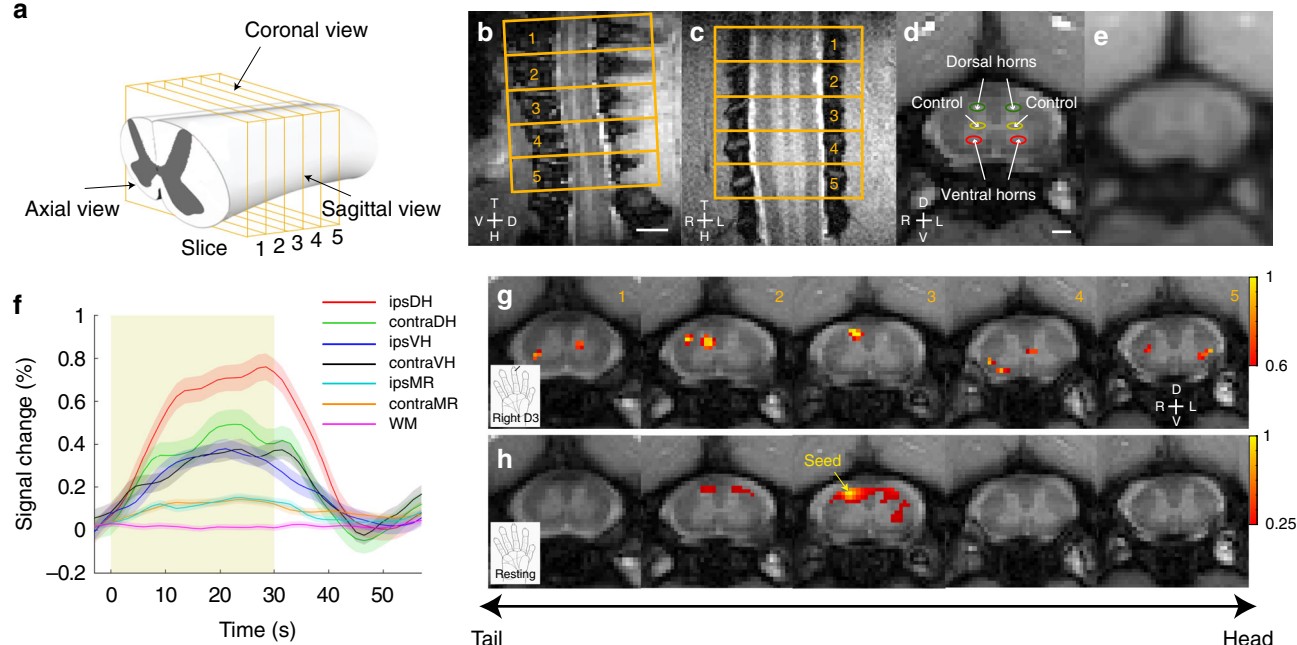

**Fig. 1** Stimulus-driven and resting-state fMRI in non-human primates at 9.4 T. **a** Schematic diagram (modified from Hollis et al.[55]) of imaging planes of the spinal cord. **b** Sagittal and (**c**) coronal views of the spinal cord in magnetization transfer contrast (MTC) images. Scale bar represents 3 mm. (**d**) MTC and (**e**) BOLD-sensitive axial images. Scale bar represents 1 mm. Red and green circles indicate dorsal and ventral horns, respectively. Yellow circles present intermediate gray matter of the spinal cord used as controls for later quantifications. **f** Group-averaged (N = 7 monkeys) BOLD signal changes in the four horns—dorsal horn (DH) and ventral horn (VH)—of the spinal cord and middle/intermediate gray matter regions (MR) that are ipsilateral (ipsi) and contralateral (contra) to the stimulus, as well as white matter (WM) control region. Shaded error bars represent standard error of mean. **g** Multirun activation map to D3 tactile stimulation thresholded at 0.6 of normalized percentage signal, with a peak value of 1. **h** Multirun resting-state connectivity patterns (thresholded at r > 0.25) of a seed from one representative monkey. D dorsal, V ventral, H head, T tail

compared with LFP connectivity patterns derived by analyzing broadband coherences between signals from different elements in each linear array. Connectivity measures between different specific pairs of ROIs were compared: dorsal–dorsal (within-slice), dorsal–intermediate gray matter, and dorsal–dorsal (across slice) (Fig. 5a). At the group level (N = 12 and 4 monkeys for fMRI and LFP, respectively), within-slice dorsal–dorsal functional connectivity was observed to be stronger (mean BOLD r = 0.49 and LFP coherence = 0.19) than that in dorsal–intermediate gray matter (averaged mean between left and right BOLD r = 0.40 and LFP coherence = 0.12) for both fMRI and LFP (Fig. 5b, c). Moreover, LFP coherence was found to be statistically significant at depths (from the dorsal surface) up to 1.5 mm between contralateral regions at precisely the same depth (Supplementary Figure 2A, B). For the four monkeys that underwent electrophysiology and fMRI recordings, linear regression between electrophysiology coherence and rsfMRI correlation revealed an r-value of 0.5079 (p = 0.0058) (Supplementary Figure 2C). The within-slice resting-state connectivity was robust between horns, but across-slice correlation strengths were found to be consistently lower (mean BOLD r = 0.26 and LFP coherence = 0.04). Trends of connectivity measures for different ROI pairs (Fig. 5b, c) were also highly similar between the two modalities and Bonferroni–Holm corrected Mann–Whitney tests between different ROI selections support this conclusion.

## Discussion
We found in this study that within-slice (representing one spinal segment) dorsal–dorsal horn resting-state functional connectivity is significantly greater than connectivity values measured between dorsal horns and reference regions, identified here as intermediate gray matter regions in the cord, for both fMRI and LFP.

Comparisons of how connectivity values varied between the three ROIs (dorsal–dorsal within-slice, dorsal–intermediate–GM, and dorsal–dorsal across the slice) between the two modalities revealed that they are highly similar. Previous spinal rsfMRI studies have also reported the existence of strongly correlated bilateral sensory networks in anesthetized rodents and NHPs[13,15], and awake humans. In humans, an early report used independent component analysis, a data-driven approach, to separate the spinal cord into dorsal and ventral networks[16]. We used an ROI-based technique that showed robust BOLD correlations between dorsal–dorsal and ventral–ventral horns, with no significant group-level correlations between gray and white matter regions[7]. The reproducibility of these findings was further quantified and confirmed[7,9], while more recent studies have explored the variable patterns of the functional network[12] and the influences of physiological noise on measurements of functional connectivity[11]. In animal studies performed at 9.4 T, similar conclusions were obtained: strong dorsal–dorsal and ventral–ventral connections were found, compared with control white matter regions[13,15]. The biophysical basis of these findings to date remains unclear for the spinal cord. Moreover, whether correlated fMRI signals between spinal ROIs reflect similar neural processes as cortical ROIs also remains largely unknown. Our results using direct electrophysiological measures help fill this knowledge gap, although our findings here focused only on the relationships of BOLD and LFPs in the sensory dorsal horns. Overall, our results demonstrate that rsfMRI can be a reliable surrogate biomarker for evaluating neural circuits in the spinal cord, and the strength of BOLD correlations reflects neural functional connectivity, as indicated by spontaneous LFP signals.

Communication between spinal segments is critical for executing spinal cord functions, but intersegment (cross-slice)

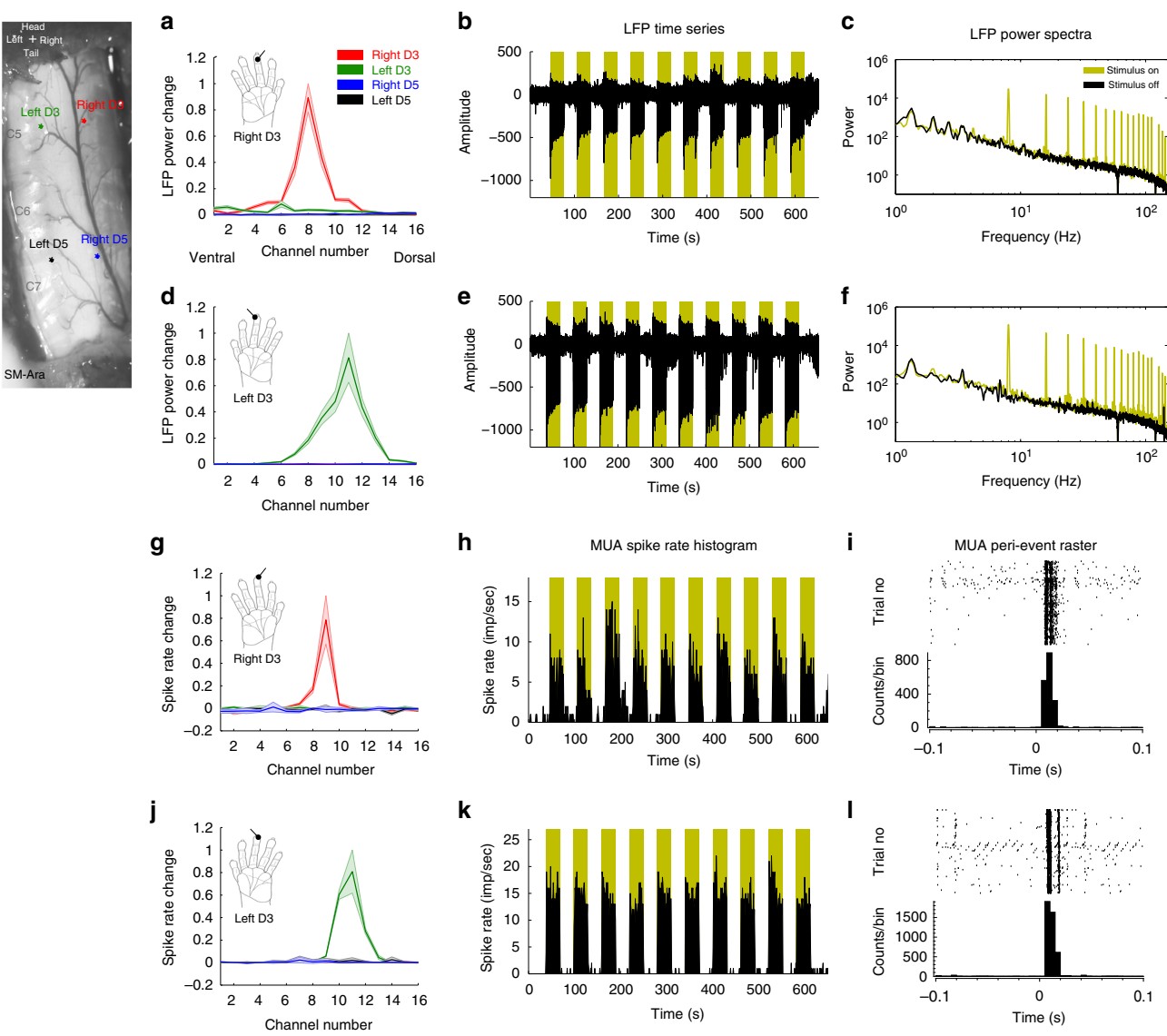

**Fig. 2** Stimulus-driven LFP and spike activity responses in one representative monkey, SM-Ara. **a**, **d** Normalized averaged ($n = 3$ runs) LFP power changes computed from Welch's power spectra, and (**g**, **j**) normalized spike rate ($n = 3$ runs) changes between 30 s stimulus-on and -off in all four electrodes under right-D3 and left-D3 innocuous 8-Hz tactile stimulus conditions, respectively. Shaded error bars represent standard error of mean. **b**, **e** LFP time series from the channel that presents the greatest power change and its respective (**c**, **f**) Welch's power spectra. Dark yellow shaded regions and plots represent stimulus-on periods. **h**, **k** Spike rate histograms at the channels corresponding to the largest changes and their corresponding (**i**, **l**) raster plots (top row) and peri-event histograms (bottom row). Spike activity was processed for only two of the four monkeys (SM-Ara and SM-Bus)

resting-sate connectivity has been little studied. In humans, Kong et al. reported no correlations between resting-state signals between different segmental levels using an independent component analysis[16]. More recently, Liu et al. showed significant functional connectivity across slices and vertebral levels, although most of them were located within one segment distance[12]. In our previous investigations in NHPs, we also found a strong functional connectivity between the same horn (e.g., dorsal to dorsal) on two adjacent segments, but with resting-state correlation values decreasing from ~r = 0.85 to ~r = 0.4 and ~r = 0.3 when moving one and two slices away, respectively[13]. A similar phenomenon was observed in rodents where connectivity decreased significantly as distances along the spine between ROIs increased[15]. With direct neuronal recordings, we aimed to answer the question of whether this observation is a possible artifact of image processing or image acquisition applied on a slice-by-slice basis, or because correlations truly decrease with increasing

separation along the cord. Our rsfMRI and electrophysiology results both indicate the latter. The diminished tactile responses two segments away from a stimulated digit's receptive field are also consistent with this finding. The segmental organization of the spinal cord permits somatotopic encoding of peripheral information, and perhaps intrinsic circuits are also organized in such a fashion. As expected, LFP measurements showed greater statistical differences between across- and within-slice connectivity ($p < 0.00005$ Bonferroni–Holm corrected Mann–Whitney test), and parallel those from fMRI.

Logothetis et al. showed, with simultaneous recordings of BOLD and neural signals in the visual cortex, that a spatially localized increase in BOLD contrast corresponds directly to an increase in local neural activity[22]. In addition, LFP responses were observed to be statistically greater and more maintained throughout a stimulus than MUA[22]. Our stimulus-driven results unequivocally showed that increased fMRI signal

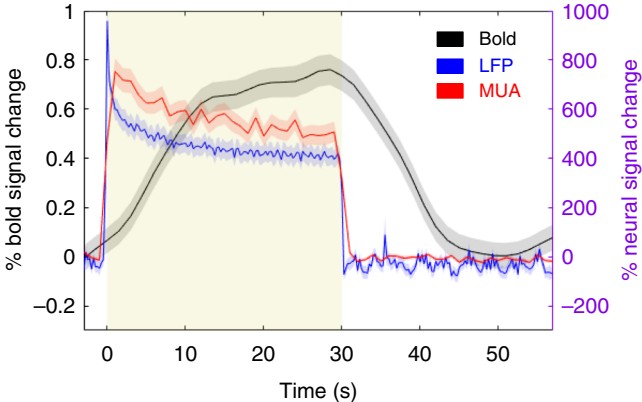

**Fig. 3** Responses to innocuous tactile stimulation of digits in the spinal cord. Overlay of averaged time courses of fMRI (black), LFP (blue), and MUA (red) responses in the ipsilateral dorsal horn. Shaded yellow region and error bars represent stimulus-on period and standard error of mean, respectively. LFP signals are represented with computed r.m.s. (bin size = 0.25 s, no overlap, $N = 4$ monkeys, $n = 40$ runs), while MUA is presented as spike rates (bin size = 1 s, no overlap, $N = 2$ monkeys, $n = 24$ runs). Percentage signal changes for BOLD and neural signal were computed relative to the pre-stimulus period; BOLD and electrophysiological signals are plotted against the y-scale on the left (black) and right (purple), respectively. It is also important to note that signals from the two modalities were not acquired simultaneously

changes correspond to increased neural electrophysiological activity. Robust LFP responses were observed in all four monkeys, and both MUA and LFP responses were maintained throughout the 30-s duration stimuli, a result similar to our observations in the cortical area 3b of squirrel monkey[32]. Moreover, our observations within the spinal cord again support the notion that regions which are engaged collectively in the same function, such as processing an external stimulus (e.g., tactile input), demonstrate a strong resting-state functional connectivity between them[6,13,33–35]. In particular, ipsilateral dorsal horns of the spinal cord have been shown to be most responsive to tactile stimulations[31,36], as further verified with both fMRI and electrophysiology in this study. In humans, innocuous tactile stimulation produces responses localized in the ipsilateral dorsal gray matter as well as in areas around the gracile and cuneate nuclei, with the overall activation pattern in line with the dorsal–medial lemniscus pathway[36]. Consistent with this observation, we also previously found tactile-evoked responses predominantly toward deeper regions of the ipsilateral dorsal horn in monkeys at 9.4 T[31]. Interestingly, responses in contralateral dorsal gray matter were also detected in both studies, and this could be attributed to a descending projection effect and/or commissural connections between bilateral dorsal horns. Using electrode positions that responded most robustly to tactile stimulation as seeds, these channels demonstrated stronger connectivity patterns with each other compared with others. Overall, these studies extend the notion that baseline functional connectivity is an organizational feature of the central nervous system and occurs in the spinal cord, and replicates patterns similar to stimulus-evoked activation foci. The observation that regions working together in tasks exhibit a strong functional connectivity at rest appears to be a universal organizational principle within the central nervous system.

While spinal electrophysiology has previously been performed in other animal models, the functional architecture and neural circuits of the central nervous system in NHPs share much greater similarities with those in humans. Previous reports have made use of NHPs to compare and understand the underlying biophysical basis of BOLD signals, but these were restricted to different cortices of the brain[37]. In this study, we have used NHPs to perform direct validation of MRI findings in the spinal cord using multichannel microelectrodes. That being said, some rsfMRI spinal connectivity differences between NHPs and humans in previously published reports are also present. In humans, within the hemi-cord, dorsal–ventral connectivity remains speculative and shows weaker correlations compared with bilateral dorsal and ventral horn connectivity[8,9,18], although unilateral resting-state dorsal components have also been identified using independent component analysis[16]. In NHPs, however, ipsilateral dorsal–ventral connectivity appears to be significant when compared with controls[13], and this could be attributed to differences in the functional organization of the spine between species if there is a greater use of coordinated bilateral behaviors in animals. Despite some differences present, the study of NHPs provides a crucial link between invasive animal data and human studies[33,38].

As described in previous publications[17,30,39,40], imaging the cervical spinal cord poses several challenges. Using an optimized MRI sequence along with modified image processing steps[13], we have reduced these confounding effects and developed a sensitive protocol for the detection of resting-state networks in the cervical spinal cord. Electrophysiological recordings of the spinal cord also pose their own challenges. For example, there is a trade-off between the number of recording sites (four were chosen in our study) and the density of electrode contacts for sampling.

Another limitation of this study is that the ventral horns were not fully sampled due to the design of the recording electrodes, which included a dead space between the tip of the electrode and the first recording contact of the linear array. Ventral–ventral connectivity has been shown to be robust and reproducible in both humans and animals[7,15,41]. In support of this finding, a number of studies have also reported the presence of commissural interneuron connections between ventral horns[42,43]. Resting-state fMRI correlations identified the corresponding U-shaped profile of connectivity that should be observable by LFP recordings using different electrode designs. In addition, the observation of hemi-cord dorsal–ventral connectivity has been less consistent. Eippert et al. reported that ROI selection appears to influence the observed dorsal–ventral connectivity, which suggests that this may be a partial volume effect that causes mixing of time courses due to the proximity of dorsal and ventral horns[44]. Given the involvement of sensorimotor systems in mediating reflexes in the spinal cord, it remains unclear why dorsal–ventral connections have not been reliably detectable. One possible explanation is that spinal cord neurons do not exhibit their full network of connections at rest[44]. Nevertheless, to fully examine the functional relationship between dorsal and ventral horns, sampling of both regions simultaneously by electrophysiology will permit more comprehensive comparisons with fMRI, but requires further developments in electrode configurations.

Achieving a high spatial resolution with adequate SNR in fMRI is critical for minimizing partial volume effects as well as for characterizing functional connectivity at fine scale. In this study, high SNR and resolution were achieved by imaging a stable preparation at 9.4 T, by acquiring images with a multi-shot gradient-echo sequence (a technique used in Barry et al., 2016, 2014)[7,45], and by reducing TE (a technique used in Zhao et al., 2009, 2008)[46,47]. Further potential improvements may be obtained using better RF coils (e.g., cryogenic coils) and deploying more advanced shimming methods.

Finally, a possible confounding factor is the influence of anesthesia in the measurement of functional connectivity for both modalities. An increase in anesthesia level has been shown to

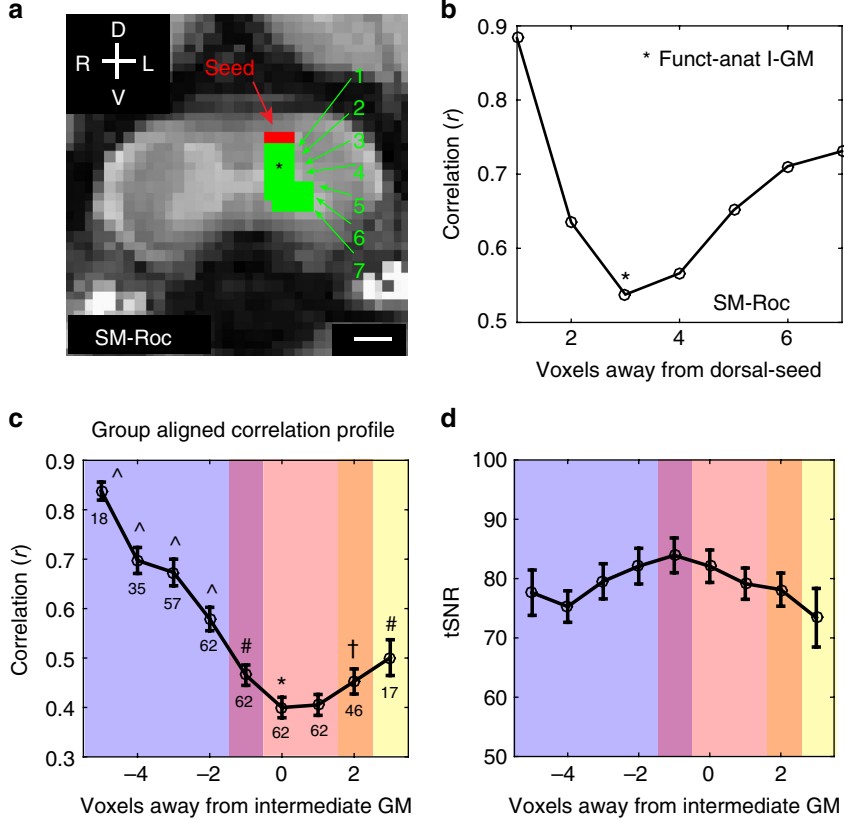

**Fig. 4** Local dorsal-to-ventral correlation profile. **a** ROIs defined in one representative monkey on one side of the spinal cord. Mean time series of the dorsal region (red voxels) was used as a seed to compute mean Pearson's cross-correlations with voxels from each depth (green voxels); depth numbers are indicated by the green numbers. Scale bar represents 1 mm. **b** An example of an observed U-shaped profile from SM-Roc. Asterisk (*) represents defined intermediate GM (I-GM) seeds based on functional and anatomical features; greatest difference of connectivity value to the dorsal horn seed. **c** Group-averaged aligned correlation profile based on each run's anatomically functionally defined seed of the intermediate GM region for both left and right sides of the spinal cord. Error bars represent standard error of mean. The number of observations for each distance are indicated below each scatter circle. Data points representing four voxels away from I-GM were removed as they contain only five observations. Blue, red, and yellow shades represent estimated dorsal, intermediate, and ventral boundaries defined based on two-sided Mann–Whitney test comparisons (corrected for false discovery rate) relative to the defined I-GM seed; ^$p < 0.000005$; #$p < 0.05$; †$p < 0.1$. Shades of overlaid colors—purple and orange—represent an estimated overlap of dorsal-IGM and ventral-IGM, respectively. Supplementary Tables 1 and 2 provide uncorrected and corrected $p$-values for Mann–Whitney Tests and $r$-values, respectively. **d** Group-averaged tSNR at different depths of the spinal cord used to compute (**c**)

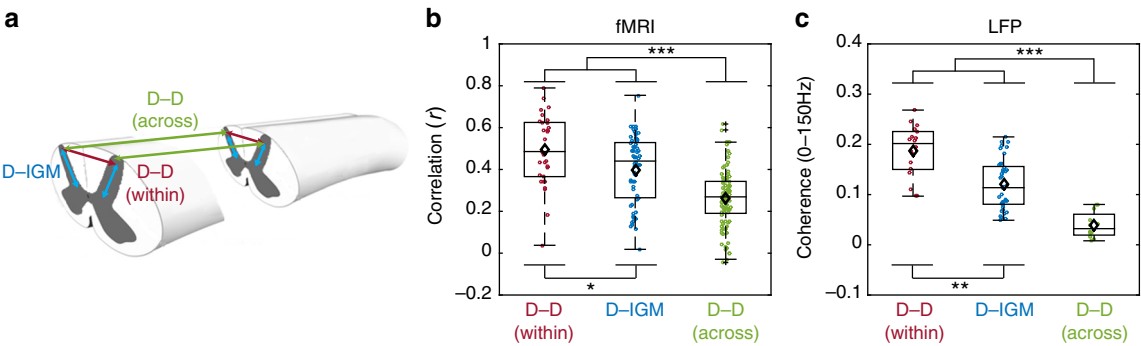

**Fig. 5** Comparison between resting-state fMRI and LFP connectivities. **a** Schematic diagram of correlation and coherences computed in the recordings of the spinal cord. **b**, **c** Group averaged within and across segments dorsal to dorsal (D–D) as well as dorsal to intermediate GM (D–IGM) connectivity was computed. Group boxplots of connectivity measures are displayed as Pearson's correlation for fMRI and averaged coherences for LFP. Each boxplot contains fMRI observations from 12 monkeys (including those that underwent electrophysiology) across 15 functional studies, and electrophysiology observations from four monkeys in panels **b** and **c**, respectively. Ipsilateral dorsal-to-intermediate GM from the left and right hemi-cord was concatenated. The median and mean are represented as horizontal lines and diamonds, respectively, in each boxplot. Scatter circles on each boxplot represent individual observations. Using mean connectivity values from the three ROI pairs from both modalities (**b**, **c**), Pearson's correlation value was computed to be 0.9982. *$p < 0.05$, **$p < 0.0005$, and ***$p < 0.00005$ Bonferroni–Holm corrected two-sided Mann–Whitney test

reduce cortical functional connectivity in NHPs[48,49], while others have indicated the possibility of neurovascular decoupling at high anesthesia dosages (see a review in Masamoto and Kanno, 2012)[50]. That being said, studies have also demonstrated that anesthesia has a weaker influence in early sensory cortical regions[51], and thus, this effect is most likely reduced in the downstream spinal cord. In our experiment, the isoflurane was maintained at <1%, which is lower than the range where significant neuronal connectivity activity drops and becomes unstable in the brain (1.75% isoflurane reported in ref. [48]). In fact, a recent study found that neurovascular coupling in the spinal cord remains unaltered at 1.2% isoflurane in an experiment with decerebrated rats undergoing electrical stimulation. Moreover, simultaneous recordings of local evoked field potentials and fMRI in brain studies have shown that signals from the two modalities are still coupled under anesthesia[22,26,52–54]. With this information, we believe that anesthesia is unlikely a major confounding factor to the trends observed in this study.

## Methods

**Animal preparation.** Adult male squirrel monkeys (Saimiri bolivians) were used in this study. Four animals (SM-Ara, SM-Leg, SM-Gua, and SM-Bus) underwent laminectomy and subsequent electrophysiological recordings (Fig. 6). For both MRI and electrophysiological experiments, animals were first sedated with keta-mine hydrochloride (10 mg/kg)/atropine sulfate (0.05 mg/kg, i.m.) and maintained with isoflurane anesthesia (0.5–1.2%) delivered in a 70:30 $N_2O/O_2$ mixture. After intubation, animals were mechanically ventilated, monitored, and infused intra-venously with 2.5% dextrose in saline solution (2–3 ml/h/kg) in order to prevent dehydration. Animals' vital signals, including peripheral oxygen saturation and heart rate, EKG, end-tidal $CO_2$, and respiratory pattern, were continuously mon-itored. With a circulating water blanket, temperatures of the animals were also kept between 37.5 and 38.5 °C. Animals were subsequently placed in a custom-designed MR cradle and onto a stereotaxic frame for MRI and electrophysiological experi-ments, respectively. Extra care and efforts were put into ensuring that the animal's neck was secured and straight while ear bars were also used to minimize further motions. While the isoflurane level fluctuated over a range depending on each animal's physiological condition, anesthesia level was generally maintained at 0.7–0.8% isoflurane during functional data acquisitions. For both MRI and elec-trophysiological experiments, animal monitoring and preparation were performed in almost identical manners. All animal procedures were in compliance and approved by the Institutional Animal Care and Use Committee of Vanderbilt University.

**MRI data acquisition and analysis.** MRI acquisitions were obtained on a 9.4 T Varian magnet with a saddle-shaped transmit–receive surface coil positioned over the neck. High-resolution anatomic axial images were obtained using magnetiza-tion transfer contrast (TR/TE = 220/3.24 ms, 0.25 × 0.25 × 3 mm³), while BOLD images were acquired using a fast gradient-echo sequence (TR/TE = 46.9/6.50 ms, matrix size = 64 × 64, field of view = 32 × 32 mm², resolution = 0.5 × 0.5 × 3 mm³, and flip angle ~15˚, ~3 s/volume). Resting-state (300 volumes) and stimulus-driven (8-Hz innocuous tactile, 30 s on/off, seven epochs) acquisitions were both obtained in 12 and 7 animals, respectively. Previously acquired spinal fMRI datasets with a higher temporal resolution (TR = 24.0 ms, ~1.5 s/volume) on healthy monkeys were included in group analyses[13,31]. We included data acquired with slightly different imaging parameters in order to increase the number of runs and animals in the group average, thereby increasing the reliability of estimates of functional connectivity strengths for comparison with electrophysiological measurements. The different timings are not expected to affect correlations. Specifically, resting-state fMRI included animals that underwent electrophysiology, while stimulus-evoked fMRI contained four monkeys from our previous study[31] and the first three animals that underwent electrophysiology (SM-Ara, SM-Leg, and SM-Gal).

Previous studies evaluated preprocessing methods for de-noising spinal fMRI data[17,30]. In this study, ROI data analyses were similar to those used in our previous publications[13,31]. Specifically, functional image volumes were first aligned using a 2D rigid body motion correction algorithm based on maximization of mutual information, with three motion parameters estimated (two translations and one rotation) slice by slice. BOLD images were then upsampled to 0.25 × 0.25 × 3 mm³ by linear interpolation to match a set of anatomical images. Manual alignment was then performed between the functional images and the corresponding structural images. This step was performed for each slice and for all the runs in each animal. For each slice, "nuisance" signals were derived from muscle and cerebrospinal fluid voxels using principal component analyses. The first three to five signal components that accounted for at least 70% of the cumulative signal variance, along with motion correction parameters, were used as signal regressors in a general linear model to mitigate their contributions to each BOLD time series. FMRI time series were then band-pass filtered with a pass band of 0.01–0.1 Hz using a Chebyshev Type II filter. Runs in which the tSNR in gray

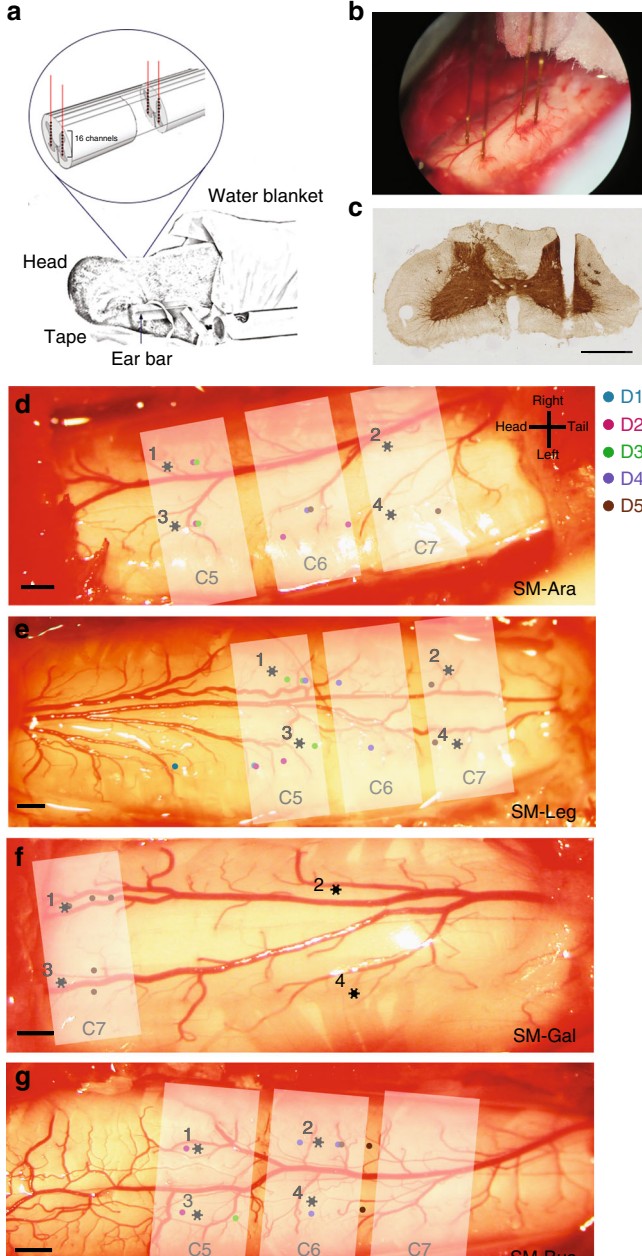

**Fig. 6** Electrophysiological recording setup, recording sites, and maps of different digit segments for the four squirrel monkeys (SMs). **a** Schematic diagram of the spinal cord inserted with four electrodes, each with 16 channels that are 150 µm apart. **b** Sample photograph of the four inserted electrodes in SM-Ara. **c** Cytochrome oxidase stain used to verify electrode penetrations of the spinal cord by identifying the electrolytic lesion. Exposed spinal cord with mapped digit regions (color dots) and recording sites (black asterisks) for (**d**) SM-Ara, (**e**) SM-Lego, (**f**) SM-Gua, and (**g**) SM-Bus. Scale bars represent 1 mm. Shaded white regions are estimated segments of the spinal cord

matter was <50 were removed from the study. In order to eliminate possible partial volume effects due to smoothing introduced by linear interpolation from functional to anatomical resolutions, a cluster threshold of 2 was applied to prevent spurious single-voxel correlations (Fig. 1g, h). Spinal cord masks, excluding voxels outside GM and WM, were also applied. Finally, ROIs were carefully defined before ROI-based analysis.

**Electrophysiological mapping and recording.** The average fMRI activation maps obtained from seven monkeys were used as guidance for microelectrode

penetrations into the exposed spinal cord. The electrophysiological experiments in this study were divided into two phases: a mapping phase, during which the response areas of the spinal cord were mapped, and subsequently a recording phase, using microelectrode arrays. During the mapping phase, single epoxylite-coated tungsten microelectrodes (~1-MΩ impedance) with standard exposed sharp tips (< 3 μm) were used. The responses of each microelectrode were recorded at different penetration depths separated by 300-μm increments. At each interval, the hand digits of the animal were tapped lightly to allow the receptive fields of neurons to be established. The magnitudes of responses were also evaluated qualitatively by listening to an audio representation of spike activity and the viewing of action potential traces. Segments of the spinal cord were then identified based on known receptive field properties as well as the somatotopic organization of digits. During the recording phase, using the maps of digit responses, four linear microelectrode arrays (Microprobes Blackrock, 16 channels, 150-μm separation between contacts) covering various depths were then carefully inserted into targeted recording sites (Fig. 6a, b). LFP broadband voltage signals and spiking activity exceeding a threshold were recorded using a Multi-channel Cerebus Neural Signal Processor system (Black Rock, Millard county, Utah), which received digitized signals at a sampling rate of 30 kHz, and a local reference. LFP signals were recorded continuously with a lowered sampling rate of 500 Hz and low pass filtered at 250 Hz. MUA was also recorded by capturing timestamps of spikes evoked using the same system. Spike processing employed a band-pass filter of 250–5000 Hz, while a global spike detection threshold was set to −4 times the root-mean-squared energy. For stimulation of each digit, 2–3 trials were obtained using the same 30-s on-/off-paradigm as for MRI with 10 epochs, while resting-state signals were recorded for a duration of 15 min.

**Stimulus-driven LFP and MUA data analysis.** LFP signals sampled at 500 Hz were notch-filtered at 60 and 120 Hz before a band-pass filter (pass band between 1 and 150 Hz) was applied. Stimulus-driven (Fig. 2) LFP data were then separated into stimulus-on and stimulus-off periods, which were identified as 20 s before and after onset times, respectively. Welch power spectra were subsequently computed (with Hamming window apodization, window length = 10 s, overlap = 50 ms). The percent power changes between stimulus-on and -off conditions at the first five harmonics of 8 Hz were calculated and normalized to the upper bound of the greatest signal. LFP recordings were analyzed for all four monkeys, while spike activity was analyzed for only two monkeys (SM-Ara and SM-Bus), because our initial electrophysiology experiments were optimized specifically for collecting LFP responses. For example, thresholds set for spike rate recordings were too high and were not ideal, resulting in minimal spike events being recorded in the initial experiments. Spike rate histograms (bin size = 1 s) and peri-event rasters (bin size = 0.005 s) were computed using NeuroExplorer software. At the group level, time series at channels with the greatest LFP responses were averaged across animals, runs, and epochs, and root-mean-square (r.m.s.) values were computed in successive 0.25-s windows, while averaged spike rate histograms (bin size = 1 s) were also calculated similarly. These plots were converted to percentage changes in spike rate or LFP signals at each timepoint during stimulus-on relative to the averaged spike rate during pre-stimulus baseline signals (3 s before the onset of a stimulus), and subsequently overlaid on the extracted fMRI BOLD activation time course (Fig. 3).

**Stimulus presentation protocol.** In order to ensure a stable contact between the probe and the animals' fingers, small pegs were glued to the fingernails of the hand, which were firmly embedded into plasticine. This setup leaves the glabrous surfaces of each digit exposed for innocuous vibrotactile stimulation by a 2-mm-diameter rounded plastic probe that is attached to a piezoelectric device driven by a Grass S48 square wave stimulator. During periods with no stimulation, the probe was in light contact with each digit. For fMRI experiments, tactile indentations (0.34 -mm vertical displacement) of the probe were presented for 30 s on followed by 30 s off (= one epoch) at 8 Hz with a pulse duration of 20 ms. In total, seven epochs were typically presented within one imaging fMRI run. The same tactile stimulus protocol was used for electrophysiological recordings with 10 epochs. Multiple runs (2–3 runs) were collected for each digit stimulation and stimulus-on and -off timestamps were recorded.

**Resting-state LFP data analysis.** Resting-state LFPs were further de-noised by notch-filtering the first five harmonics of the respiration frequency. Dorsal seeds at each shank of the electrode that were responsive to tactile stimulation were identified as channels with the greatest LFP responses. Recordings from electrodes 2 and 4 of SM-Bus were excluded due to residual contamination from respiratory and cardiac motion. Intermediate gray matter, which serves as a reference tissue, was subsequently identified as the channel closest to 1 mm away from the dorsal seed on the same shank. This measurement was made based on the corresponding high-resolution MRI axial images. Resting-state magnitude-squared coherences, which are a function of the power and cross-power spectral densities of two signals, were then computed between dorsal horns within and across segments and used to quantify the functional connectivity between the regions. Dorsal-to-intermediate gray matter coherences were also measured within each segment. Similarly, functional connectivity with the contralateral side of the spinal cord was computed using coherences between channels distant from the dorsal seeds on each

shank. Because penetration depths of each monkey were slightly different, measurements that contained data from only one monkey were not considered for group analyses.

**ROI selection of intermediate GM.** ROI voxels in the intermediate GM control regions were selected based on both anatomical features and functional patterns relative to the ipsilateral dorsal horn. Specifically, small ROIs (2–5 voxels wide (i.e., across the cord)) × 1 voxel deep (i.e., in a dorsal–ventral direction)) were drawn from dorsal to ventral horn on one side of the spinal cord (Fig. 4a). The dorsal horn (red voxels) was subsequently selected as the seed. Mean correlations between resting-state time series were then computed for all the ROIs as a function of depth (green). The correlations demonstrated a U-shaped pattern in which correlation strengths decreased monotonically from a high value to a low value approaching the intermediate region, and then increased again as the ventral region was approached (Fig. 4b). We then functionally and anatomically defined the most appropriate intermediate GM seeds as those voxels at the depth that showed the greatest difference in correlation from the dorsal seed (typically 4–5 voxels deeper). Given the unclear anatomic boundaries of the intermediate GM region, we believe that this allowed a more precise identification of a control intermediate region. Our results showed that a method of defining the intermediate GM region through anatomical features alone may encompass a region contaminated by functional signals from the dorsal or ventral horns, and thus may misplace the most appropriate location of the control region. For sessions in which not all runs showed this profile, the intermediate GM region was defined using results from those runs which were U-shaped. Otherwise, they were identified anatomically as voxels at the same depth as the central canal of the spinal cord. This was repeated for all runs for both left and right sides of the spinal cord. This pattern was supported at the group level with aligned correlation profiles at the defined intermediate–GM ROI shown in Fig. 4c.

**ROI-based and statistical analyses.** An ROI-based correlation analysis was used to assess functional connectivity between different ROIs. This method was selected because the ROIs could be drawn based on stimulus-driven data as well as previous fMRI studies in humans and animals. We have previously shown that sensory evoked responses are located in the dorsal and ventral horns[31], so here we evaluate how specific ROIs in dorsal and ventral horns are interconnected in a resting state. Dorsal ROIs were manually selected (2–3 voxels) using co-registered high-resolution MTC-weighted images that showed a high contrast between the white and gray matter butterfly. This was done for both the left and right side of the spinal cord for each monkey. Pearson's correlation coefficients were subsequently computed between different ROI pairs of interest. Our quantitative connectivity measurements between ROIs were similar to those described in our previous publications[7,8,13,15]. Briefly, mean time series of $m$ individual voxel time series of an ROI (either dorsal or ventral horn of the spinal cord) were correlated with $n$ time series of another ROI. Subsequently, the average of this correlation vector was selected as the metric of functional connectivity between the two ROIs. This was performed for different pairs of ROIs of interest within (middle slice, slice 3 of Fig. 1) and across slices (slices 2, 3, and 4 of Fig. 1). Statistical tests of significance were performed between correlation values using nonparametric two-sided Mann–Whitney Wilcoxon tests. Bonferroni–Holm corrections were performed, and $p < 0.05$ was considered to be statistically significant.

**Histology.** At the conclusion of invasive electrophysiological recording sessions, animals were given a lethal dose of anesthetic (sodium pentobarbital) before being perfused transcardially with phosphate-buffered 0.9% saline, then 2–4% paraformaldehyde in phosphate buffer, and last 10% sucrose in phosphate buffer. For one of the monkeys (SM-Gua), an electrolytic lesion was made by passing a current (10 μA) via an electrode in one of the recording sites in the spinal cord (Fig. 6c). This lesion was made to further confirm our sampling region of the spinal cord. The spinal cord was subsequently extracted before sections of the spinal cord were stained for cytochrome oxidase to locate electrode penetration.

## Data availability
A reporting summary for this article is available as a Supplementary Information file. The source data underlying Figs. 1F, 2, 3, 4, and 5 are provided as Source Data files at https://figshare.com/s/830745f89f8d23777867. Data and codes supporting the conclusions presented here will be made available following reasonable requests to authors.

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

## Acknowledgements

This study is supported by the NIH grant NS092961 and the DOD grant SC160154. The authors gratefully acknowledge Fuxue Xin, George Wilson III, Dr. Qing Liu, and Chaohui Tang for their assistance with data collection. We also thank Dr. Chia-Chi Liao for advice on spinal cord recordings, and Dr. Robert L. Barry, Dr. Baxter Rogers, and Benjamin Conrad for their advice on spinal functional data analysis.

## Author contributions

J.C.G. and L.M.C. designed and supervised the research; T.-L.W., P.-F.Y., Z.S., R.W., F.W., and L.M.C. performed the animal studies; T.-L.W., P.-F.Y., and A.M. contributed new analytical tools and analyzed the data; and T.-L.W., L.M.C., and J.C.G. wrote the paper.

## Additional information

**Competing interests:** The authors declare no competing interests.

