## [Transparent Peer Review File · Nature Communications]

Reviewers' comments:

Reviewer #1 (Remarks to the Author):

NCOMMS-18-30916

The paper describes the application of sensory (digit) stimulation in the squirrel monkey to record spinal activity either via electrophysiological techniques or by using functional MRI at high field (9.4T). The sites of activity were then further explored through analysis of data acquired at rest. The ultimate goal was to relate the two measures and demonstrate whether the patterns of resting state correlation, previously shown by the authors, have an underlying neuronal basis. Furthermore, the authors address whether connectivity was greatest between responding neurons at the bilateral segmental level, or if it could also be observed across segments. This is an elegant study, but I would like to see some further justification for the assertion that the ephys and fMRI data are showing the same patterns of correlated activity. Should the patterns of correlation (with both ephys and fMRI) be shown to be reproducible, then the presented data would provide a sound basis for future work examining spinal resting state connectivity.

Major points:

Not enough details about how the comparison was made between the connectivity patterns determined with fMRI and ephys. What are the "trends of connectivity measures" (Page 12, Line 268) and is an r value of 0.9982 really plausible?

What consideration was given to the problem of multiple comparisons when examining the U-shaped pattern of connectivity as you traversed from dorsal to ventral cord?

To what extent are the presented data reproducible?

Less major points:

At various points in the manuscript it was not clear which animals were being used for which part of the experiments, or whether some animals underwent both ephys and MRI.

(Page 3, Line 30) Please add Kong et al., 2014 to the list of studies demonstrating FC in the spinal cord, as this appeared at the same time as the report by Barry et al., 2014 - so should be given equal prominence.

(Page 3, Line 37): "led us to hypothesize that" - these hypotheses were clearly (also) stated in papers originating outside of the authors' own lab. This could be written in a more even-handed fashion.

(Page 3, Line 50): "yet no study has directly compared and related rsfMRI signal to spontaneous neural activity within the spinal gray matter" - not strictly true. Brieu and colleagues recorded blood flow changes with an optical technique, comparing responses to those recorded electrically from the rat spinal cord. Whilst that study relates to stimulus related activity rather than that at rest, I think it is beholden on the authors to at least cite in the Introduction this seminal study as it provides motivation for the current work.

Brieu, N., Beaumont, E., Dubeau, S., Cohen-Adad, J., & Lesage, F. (2010). Characterization of the hemodynamic response in the rat lumbar spinal cord using intrinsic optical imaging and laser speckle. *Journal of Neuroscience Methods*, 191(2), 151–157.

(Page 4, Line 78). Point of clarity - were ephys animals used for MRI? Or was it just the remaining 8 monkeys? If so, why were data reported only for 7 animals (Figure 2 legend)?

(Page 5, Line 92). What was the rationale for including data acquired under different imaging conditions?

(Page 5, Line 100). Missing the word "were".

(Page 6, Line 112). Point of confusion - "For each animal" - is this for the 4 animals with the laminectomy? If so, this can be written much more clearly i.e. group activation from the 8(?) monkeys was used to guide positioning of microelectrodes....?

(Page 6, Line 115). Point of confusion - "penetration depths were recorded and performed at 300micrometre increments". Why was this necessary as in Figure 1 you have 16 contact sites spanning the dorso-ventral extent of the cord? The way it currently reads is that you pushed the electrode in to different depths then made your recordings, but I thought this would be unnecessary given the experimental set up? Apologies if I have misunderstood this.

Okay I understand now! You have a search phase (please emphasise this in the paper) and then once your location has been mapped, a micro-electrode array recording phase.

(Page 8, Line 175). Consistency - you refer to the correlations as being between the seed and other ROIs as a function of depth, whilst in the Figure legend you refer to the correlations as being between the seed and "layers". Please avoid using the description layers throughout the manuscript, as the presented imaging data do not allow such assignment.

(Page 9, Line 188). At the group level, how many animals contributed? For the small number of animals studied, the data should be presented using standard deviation (rather than standard errors). Similarly, was a consideration made for the number of statistical tests performed? E.g. how do the number of observations (Figure 5C) relate to the number of animals? Should the reader be worried about potential false positives?

(Page 14, Line 334). I think it is necessary for the authors to be clear about what they can and cannot resolve with their imaging, using the appropriate evidence to justify their findings. E.g. the use of nociceptive stimulation (here referred to as "pain"), which may give rise to both superficial and deeper activity in lamina V of the dorsal horn, cannot really be used to justify the observed patterns of ephys or fMRI activity in response to a vibrotactile stimulus. I would also like to see the relevant spinal ephys literature and expected location of activity (for the stimulus used) related to the patterns of observed activation.

Reviewer #2 (Remarks to the Author):

The study compares resting-state neural signaling, and also activity evoked by innocuous touch, in the NHP spinal cord, measured by electrophysiology and fMRI. Electrophysiology was carried out in 4 monkeys and 12 monkeys were studied with fMRI.

Functional MRI data were acquired at 9.4 T using previously-established methods for NHP studies. The data quality appears to be outstanding.

The results demonstrate a correspondence between BOLD fMRI signal changes, and LFP and MUA recorded with electrophysiology, in response to innocuous tactile (vibration) stimulation on one digit. The results also show consistent connectivity in the resting-state, as measured by fMRI and electrophysiological methods.

The robustness of the findings would be increased if the authors could show that the connectivity measured in the resting-state is distinct from that measured during innocuous stimulation. It is puzzling that given the analysis done with the task data, that the resting-state connectivity was

not assessed between dorsal and ventral regions.

The results also demonstrated correlation of measured signals (connectivity) between the superficial dorsal horn and the ventral horn, with lower connectivity to intermediate gray matter. This provides strong evidence of the sensitivity of the results and goes a long way to eliminating the possibility that correlations were artifacts of correlated noise (random or physiological) in the data. The results are thus very convincing.

Overall this paper is extremely well written and the results are clearly presented, with the exception of some minor challenges with clarity as detailed below. The methods appear to be very robust and the results are convincing. The results are important because they can be expected to impact on the use of fMRI in the spinal cord and contribute to better understanding of spinal cord functional organization, while also helping to establish the presence of coordinated resting-state activity in the spinal cord.

I have only one major concern (as already mentioned above): why didn't the authors compare resting-state functional connectivity between dorsal and ventral regions?

The remaining comments are minor points (some very minor), primarily concerning the clarity of the manuscript:

- 1) "Data pre-processing and ROI analysis procedures of spinal data were standard ..."
(how can the authors call this method "standard" if it has only been used in two previous studies?)
- 2) "fMRI images" - the "f" is redundant
- 3) Line 100/101: "For each slice, "nuisance" signals derived from muscle and cerebrospinal fluid voxels using principal component analysis."
This sentence is unclear. Is there a word missing?
- 4) In the section "MRI Data Acquisition and Analysis", the previous published work that established these pre-processing methods for spinal cord fMRI should be cited. There are only two papers cited in this section.
- 5) Line 165 (also line 167): "...coherences were also measured..." This seems like vague/strange wording. Do the authors mean specifically correlation between the measured signals?
- 6) Lines 191-193 (with references removed for clarity): "This method was selected, as were in our previous spinal studies, because ROIs drawn could be made based on hypothesis derived from stimulus-driven data as well as previous fMRI studies in humans and animals."
This sentence is not clear at all.
- 7) Line 194: should the word "aligned" be replaced with "co-registered", to make the sentence more clear? (If not, then this sentence is not clear).
- 8) Line 197: "...horn-horn connectivity measurements..." This is jargon and should be written in a more clear form. Similarly, several times in the sentence that follow the authors simply refer to "horns". This is a somewhat sloppy/jargon way to write.
- 9) Line 278: "We found in this study that within-slice (representing one spinal segment) dorsal-dorsal horn resting-state functional connectivity is significantly greater than values for dorsal horn to reference regions, ..."
This phrase would be more clear written as "We found in this study that within-slice (representing one spinal segment) dorsal-dorsal horn resting-state functional connectivity is significantly greater than connectivity values measured between dorsal horn and reference regions, ..."

10) Line 280: "Quantitative comparisons of connectivity trends ..."

I don't believe the authors actually quantified the connectivity trends. This sentence is unclear.

11) The paper cited to support the statement that "imaging the cervical spinal cord poses several challenges" is given as Eippert et al., 2016a. This is a very odd choice. There are many papers demonstrating these challenges, and methods to mitigate them, going back many years. A recent review article has summarized them: Powers et al. 2018 (<https://www.ncbi.nlm.nih.gov/pubmed/30201938>)

Reviewer #3 (Remarks to the Author):

Wu et al., investigated spinal gray matter functional connectivity via both depth electrophysiological recordings and fMRI in monkeys to validate rsfMRI for assessing spinal FC and to claim basic intrinsic functional architecture in spinal neuronal activity in resting state. By comparison of rsfMRI to LFP and MUA signals across dorsal-intermediate-ventral and adjacent segments along spinal cord based on ROI method, a well agreement observed in spatial connectivity pattern. The comparison conducted on both sensory stimulation and analysis of spontaneous activity in these regions. The comparison experimental design is strict. Relative to brain fMRI, spinal segment may be much smaller and challenge to have high spatial information on tiny gray matter. For example, along the gray matter of dorsal to ventral in imaging plane, only few EPI voxels, so partial volume effects may be a potential issue for precision of mapping functional activity, but ROI may be better alternative to choose as authors did. However, ROI analysis may lose detailed spatial information on functional architecture.

line 89/90, TR=46.9ms, ~3s/volume, some details need to be clear on the fMRI scan.

line 112, LFP and MUA recording parameters need details.

figure 4 showing % neural signal changes, how did you calculate the percentage for LFP and MUA?

line 245, the conclusion of U-shaped pattern can not be exactly drew without clear functional mapping, or just based on few ROI analysis.

The fMRI and ephys were not simultaneously obtained, some details needed on how well these experiments were controlled in same physiological conditions and anesthesia states.

The ephys recording and fMRI scans were conducted under isoflurane, certainly anesthesia effects may present for functional connectivity. Related discussion, especially refer to simultaneous LFP/fMRI animal brain studies, will help.

The limitations and challenges may include fMRI spatial resolution due to tiny spinal gray matter.

Potential improvement in discussion may be appreciated.

RESPONSES TO REVIEWERS

Reviewer #1 (Remarks to the Author):

NCOMMS-18-30916

The paper describes the application of sensory (digit) stimulation in the squirrel monkey to record spinal activity either via electrophysiological techniques or by using functional MRI at high field (9.4T). The sites of activity were then further explored through analysis of data acquired at rest. The ultimate goal was to relate the two measures and demonstrate whether the patterns of resting state correlation, previously shown by the authors, have an underlying neuronal basis. Furthermore, the authors address whether connectivity was greatest between responding neurons at the bilateral segmental level, or if it could also be observed across segments. This is an elegant study, but I would like to see some further justification for the assertion that the ephys and fMRI data are showing the same patterns of correlated activity. Should the patterns of correlation (with both ephys and fMRI) be shown to be reproducible, then the presented data would provide a sound basis for future work examining spinal resting state connectivity.

Major points:

Not enough details about how the comparison was made between the connectivity patterns determined with fMRI and ephys. What are the “trends of connectivity measures” (Page 12, Line 268) and is an r value of 0.9982 really plausible?

We compared the differences in mean correlation values among the three pairs of regions of interests (ROIs) (dorsal-dorsal within slice, dorsal-intermediate-gray-matter, dorsal-dorsal across slice) measured with rsfMRI (three boxplot means in Figure 6B) versus LFP (three boxplot means in Figure 6C). For both measures, the correlation between dorsal-to-dorsal (D-D) horns within the same image slice (D-D within) was the strongest whereas the correlations between dorsal-to-dorsal horns on different image slices (D-D, across) were the weakest. The decreasing pattern of correlation strengths from D-D within to D-IGM to D-D across was very similar and exhibited a high correlation r -value of 0.9982. What this high correlation indicates is that both rsFC and LFP were able to identify the differences in connectivity strength between ROI pairs in a very comparable manner. We have now clarified and moved this correlation value to Figure 6's legend. For the four monkeys that underwent electrophysiology and fMRI recordings, we performed an additional linear regression between electrophysiology coherence and rsfMRI correlation, which revealed a r -value of 0.5079 ($p=0.0058$). This is now also included in Figure 6-1C and indicated on lines 291-293 of the manuscript.

What consideration was given to the problem of multiple comparisons when examining the U-shaped pattern of connectivity as you traversed from dorsal to ventral cord?

We did not consider multiple comparisons when examining the U-shaped pattern in our initial submission but now have incorporated this at the suggestion of the reviewer. Specifically, in Figure 5C, we performed a false discovery rate (FDR) correction for the number of statistical comparisons made with the group of connectivity values defined at intermediate GM. The statistical p -values of comparisons after correction did not alter our results. Specific p -values of comparisons before and after correction are presented here in the table below:

Table#1: P-values from the Wilcoxon ranksum tests performed in Figure 5C before and after FDR correction

	Voxels away from intermediate GM								
	-5	-4	-3	-2	-1	0	1	2	3
p-value (uncorrected)	1.67×10^{-10}	1.27×10^{-11}	4.62×10^{-11}	2.68×10^{-7}	2.70×10^{-2}	N/A	7.55×10^{-1}	8.14×10^{-2}	2.53×10^{-2}
p-values (FDR-corrected)	5.02×10^{-10}	1.14×10^{-10}	2.08×10^{-10}	6.02×10^{-7}	4.05×10^{-2}	N/A	7.55×10^{-1}	9.15×10^{-2}	4.05×10^{-2}

We have updated Figure 5's legend to indicate that the p-values were FDR corrected after making the symbol ^ represent $p < 0.000005$ instead of $p < 0.0000005$; the remaining symbols and their corresponding p-value ranges remain valid after FDR correction.

To what extent are the presented data reproducible?

The reproducibility of resting-state functional connectivity in humans using fMRI between bilateral dorsal and ventral horns have previously been examined (Barry et al., 2016; Eippert et al., 2017; Wu et al., 2017), and appears to be reproducible across laboratories (see references in lines 30-32). In this current study, functional connectivity using fMRI are consistent with what was observed in previous studies in humans and primates. We believe measures of the reproducibility of the more novel electrophysiology datasets (LFP coherences) are inherent in the boxplots presented in Figure 6C as all data points across the four monkeys were overlaid on the boxplots.

Less major points:

At various points in the manuscript it was not clear which animals were being used for which part of the experiments, or whether some animals underwent both ephys and MRI.

In response, we have now clarified which animals contributed to what measurements in "MRI Data Acquisition and Analysis" under Materials and Methods.

(Page 3, Line 30) Please add Kong et al., 2014 to the list of studies demonstrating FC in the spinal cord, as this appeared at the same time as the report by Barry et al., 2014 - so should be given equal prominence.

Yes, we agree and have added (Kong et al., 2014) to the list of studies.

(Page 3, Line 37): "led us to hypothesize that" - these hypotheses were clearly (also) stated in papers originating outside of the authors' own lab. This could be written in a more even-handed fashion.

We have now reworded this statement: "These observations led several research groups to hypothesize that, like the brain, spinal cord gray matter exhibits its own intrinsic functional architecture that serves as a fundamental framework for executing and maintaining sensory, motor and autonomic functions."

(Page 3, Line 50): “yet no study has direct compared and related rsfMRI signal to spontaneous neural activity within the spinal gray matter” - not strictly true. Brieu and colleagues recorded blood flow changes with an optical technique, comparing responses to those recorded electrically from the rat spinal cord. Whilst that study relates to stimulus related activity rather than that at rest, I think it is beholden on the authors to at least cite in the Introduction this seminal study as it provides motivation for the current work.

Brieu, N., Beaumont, E., Dubeau, S., Cohen-Adad, J., & Lesage, F. (2010). Characterization of the hemodynamic response in the rat lumbar spinal cord using intrinsic optical imaging and laser speckle. *Journal of Neuroscience Methods*, 191(2), 151–157.

We agree and would like to thank to reviewer for pointing us to this study. We have now revised the statement and included this study in the Introduction on lines 50-54 of the revised manuscript (extracted below).

“In particular, Brieu et al. measured changes in blood volume and flow using intrinsic optical imaging following electrical stimulation, and subsequently compared optical responses with electrophysiological measurements (Brieu et al., 2010). While this study provides complementary insights to the work presented here, no previous study has directly compared and related rsfMRI signals to neural activity within the spinal gray matter in a resting state.”

(Page 4, Line 78). Point of clarity - were ephys animals used for MRI? Or was it just the remaining 8 monkeys? If so, why were data reported only for 7 animals (Figure 2 legend)?

We apologize for the confusion here. For stimulus-driven data presented in Figure 2, there were 7 monkeys involved: 4 monkeys from our previous study (Yang et al., 2015) and the remaining 3 monkeys were from the first 3 monkeys that underwent electrophysiology (SM-Ara, SM-Leg, SM-Gal). We have now conveyed this information to the readers on lines 102-104 of the revised manuscript. While we could include additional stimulus-driven monkey experiments, the characterization of stimulus-evoked responses has already been reported (Yang et al., 2015) and we believe does not present novel information with the addition of more animals to the group. As for resting-state fMRI studies (Figure 5 and Figure 6), runs from 12 animals, including the four monkeys that underwent electrophysiology, were included. This information has now been indicated on the figure legend of Figure 6; Figure 5-1 in Supplementary Information also reveals all the animals used.

(Page 5, Line 92). What was the rationale for including data acquired under different imaging conditions?

Given the limited number of runs acquired from each monkey, the rationale to include data acquired under different imaging conditions (volume acquisition time of ~1.5s and ~3s) was to increase the number of runs and animals to the group in order to more reliably estimate functional connectivity strengths for comparison with electrophysiological measurements. The different scan timings are not expected to affect correlations. This rationale has now been added to the revised manuscript on lines 98-101.

(Page 5, Line 100). Missing the word “were”.

The word “were” is now added and reflected on line 113 of the revised manuscript.

(Page 6, Line 112). Point of confusion - "For each animal" - is this for the 4 animals with the laminectomy? If so, this can be written much more clearly i.e. group activation from the 8(7?) monkeys was used to guide positioning of microelectrodes....?

Yes, group activation from the 7 monkeys was used to guide positioning of microelectrodes. Thank you for pointing this out and we have updated this on lines 124-125 of the revised manuscript.

(Page 6, Line 115). Point of confusion - "penetration depths were recorded and performed at 300micrometre increments". Why was this necessary as in Figure 1 you have 16 contact sites spanning the dorso-ventral extent of the cord? The way it currently reads is that you pushed the electrode in to different depths then made your recordings, but I thought this would be unnecessary given the experimental set up? Apologies if I have misunderstood this.

Okay I understand now! You have a search phase (please emphasise this in the paper) and then once your location has been mapped, a micro-electrode array recording phase.

Yes, that is correct! We apologize for not making this clear the first time around. We have now emphasized this by specifically indicating our electrophysiological recording involves two phases (a mapping phase and a recording phase) in lines 125-128 of the revised manuscript.

(Page 8, Line 175). Consistency - you refer to the correlations as being between the seed and other ROIs as a function of depth, whilst in the Figure legend you refer to the correlations as being between the seed and "layers". Please avoid using the description layers throughout the manuscript, as the presented imaging data do not allow such assignment.

We agree and have now replaced the word "layer(s)" with "depth(s)" throughout the manuscript.

(Page 9, Line 188). At the group level, how many animals contributed? For the small number of animals studied, the data should be presented using standard deviation (rather than standard errors). Similarly, was a consideration made for the number of statistical tests performed? E.g. how do the number of observations (Figure 5C) relate to the number of animals? Should the reader be worried about potential false positives?

At the group level, a total of 12 animals (Figures 5 and 6) contributed. For Figure 5C, the number of runs for each point are indicated at the bottom of each scatter point, and the number of animals is also now indicated on Figure 6's figure legend. While the number of the animals may be small, the total number of runs presented is relatively large for justifying the use of standard errors on the plots. In regard to the possibility of false positives in Figure 5C, we now have performed FDR corrections as we addressed in a previous comment. P-values were also corrected for in Figure 6.

(Page 14, Line 334). I think it is necessary for the authors to be clear about what they can and cannot resolve with their imaging, using the appropriate evidence to justify their findings. E.g. the use of nociceptive stimulation (here referred to as "pain"), which may give rise to both superficial and deeper activity in lamina V of the dorsal horn, cannot really be used to justify the observed patterns of ephys or fMRI activity in response to a vibrotactile stimulus. I would also like to see the relevant spinal ephys literature and expected location of activity (for the stimulus used) related to the patterns of observed activation.

We agree and are aware of the differential activation patterns observed between nociceptive and innocuous tactile stimulations. Our initial intention was to convey that stimulus-evoked patterns (either innocuous or noxious) have been shown to be predominant in the ipsilateral dorsal horn, and that seeds defined in these regions also exhibit strong resting-state functional connectivity. In fact, these observations are indeed consistent with what was observed in this study. That being said, we do agree that such generic comparison with noxious stimulation may not be appropriate and crude to justify our observation. Thus, we have now removed mentions or references that have made use of nociceptive stimulation evidences.

To clarify this point in the manuscript, we have appended the following additional information on the location of activation for tactile stimulus in the revised manuscript in lines 361-368:

“In humans, innocuous tactile stimulation produces responses localized in the ipsilateral dorsal gray matter as well as in areas around the gracile and cuneate nuclei, with the overall activation pattern in line with the dorsal-medial lemniscus pathway (Ghazni et al., 2010). Consistent with this observation, we also previously found tactile-evoked responses predominantly towards deeper regions of the ipsilateral dorsal horn in monkeys at 9.4T (Yang et al., 2015). Interestingly, responses in contralateral dorsal gray matter were also detected in both studies, and this could be attributed to a descending projection effect and/or commissural connections between bilateral dorsal horns.”

References for Reviewer#1

- Barry, R.L., Rogers, B.P., Smith, S.A., Gore, J.C., 2016. Reproducibility of resting state spinal cord networks at 7 Tesla. *Neuroimage* 133, 31–40. <https://doi.org/10.1016/j.neuroimage.2016.02.058>
- Brieu, N., Beaumont, E., Dubeau, S., Cohen-Adad, J., Lesage, F., 2010. Characterization of the hemodynamic response in the rat lumbar spinal cord using intrinsic optical imaging and laser speckle. *J. Neurosci. Methods*. <https://doi.org/10.1016/j.jneumeth.2010.06.012>
- Eippert, F., Kong, Y., Winkler, A.M., Andersson, J.L., Finsterbusch, J., Büchel, C., Brooks, J.C.W., Tracey, I., 2017. Investigating resting-state functional connectivity in the cervical spinal cord at 3 T. *Neuroimage* 147, 589–601. <https://doi.org/10.1016/j.neuroimage.2016.12.072>
- Ghazni, N.F., Cahill, C.M., Stroman, P.W., 2010. Tactile sensory and pain networks in the human spinal cord and brain stem mapped by means of functional MR imaging. *Am. J. Neuroradiol.* <https://doi.org/10.3174/ajnr.A1909>
- Kong, Y., Eippert, F., Beckmann, C.F., Andersson, J., Finsterbusch, J., Buchel, C., Tracey, I., Brooks, J.C., 2014. Intrinsically organized resting state networks in the human spinal cord. *Proc Natl Acad Sci U S A* 111, 18067–18072. <https://doi.org/10.1073/pnas.1414293111>
- Wu, T.-L., Wang, F., Mishra, A., Wilson, G.H., Byun, N., Chen, L.M., Gore, J.C., 2017. Resting-state functional connectivity in the rat cervical spinal cord at 9.4 T. *Magn. Reson. Med.* <https://doi.org/10.1002/mrm.26905>
- Yang, P.-F., Wang, F., Chen, L.M., 2015. Differential fMRI Activation Patterns to Noxious Heat and Tactile Stimuli in the Primate Spinal Cord. *J. Neurosci.* 35, 10493–10502. <https://doi.org/10.1523/JNEUROSCI.0583-15.2015>

Reviewer #2 (Remarks to the Author):

The study compares resting-state neural signaling, and also activity evoked by innocuous touch, in the NHP spinal cord, measured by electrophysiology and fMRI. Electrophysiology was carried out in 4 monkeys and 12 monkeys were studied with fMRI.

Functional MRI data were acquired at 9.4 T using previously-established methods for NHP studies. The data quality appears to be outstanding.

The results demonstrate a correspondence between BOLD fMRI signal changes, and LFP and MUA recorded with electrophysiology, in response to innocuous tactile (vibration) stimulation on one digit. The results also show consistent connectivity in the resting-state, as measured by fMRI and electrophysiological methods.

The robustness of the findings would be increased if the authors could show that the connectivity measured in the resting-state is distinct from that measured during innocuous stimulation. It is puzzling that given the analysis done with the task data, that the resting-state connectivity was not assessed between dorsal and ventral regions.

The results also demonstrated correlation of measured signals (connectivity) between the superficial dorsal horn and the ventral horn, with lower connectivity to intermediate gray matter. This provides strong evidence of the sensitivity of the results and goes a long way to eliminating the possibility that correlations were artifacts of correlated noise (random or physiological) in the data. The results are thus very convincing.

Overall this paper is extremely well written and the results are clearly presented, with the exception of some minor challenges with clarity as detailed below. The methods appear to be very robust and the results are convincing. The results are important because they can be expected to impact on the use of fMRI in the spinal cord and contribute to better understanding of spinal cord functional organization, while also helping to establish the presence of coordinated resting-state activity in the spinal cord.

I have only one major concern (as already mentioned above): why didn't the authors compare resting-state functional connectivity between dorsal and ventral regions?

We would like to thank the reviewer for pointing this out. We agree and recognize the importance and impact of sampling both dorsal and ventral regions in our electrophysiology experiments. However, as pointed out in our initial submission, the restricted sampling field-of-view (due to the design of the recording electrodes, which included a dead-space between the tip of the electrode and the first recording contact of the linear array) prevented ventral horns to be fully sampled. Specifically, the tungsten tip of the electrode does not contain any contacts for recordings, and the penetration depth into the spinal cord is limited given the presence of ventral vertebrae bone. We realize that this was not made clear in our initial submission and thus, have now included a separate paragraph in lines 401-417 that highlights the importance of sampling the ventral regions and indicate why this was not performed:

“Another limitation of this study is that the ventral horns were not fully sampled due to the design of the recording electrodes, which included a dead-space between the tip of the electrode and the first recording contact of the linear array. Ventral-ventral connectivity has been shown to be robust and reproducible in both humans and animals (Barry et al., 2018, 2014; Wu et al., 2017). In support of this finding, a number of studies have also reported the presence of commissural interneuron connections between ventral horns (Bannatyne et al., 2003; Jankowska, 2008). Resting state fMRI correlations identified the corresponding U-shaped profile of connectivity that should be observable by LFP recordings using different electrode designs. In addition, the observation of hemi-cord dorsal-ventral connectivity has been less consistent. Eippert et al. reported that ROI selection appears to influence the observed dorsal-ventral connectivity, which

suggests this may be a partial volume effect that causes mixing of time courses due to the proximity of dorsal and ventral horns. Given the involvement of sensorimotor systems in mediating reflexes in the spinal cord, it remains unclear why dorsal-ventral connections have not been reliably detectable. One possible explanation is that spinal cord neurons do not exhibit their full network of connections at rest (Eippert et al., 2017). Nevertheless, to fully examine the functional relationship between dorsal and ventral horns, sampling of both regions simultaneously by electrophysiology will permit more comprehensive comparisons with fMRI but requires further developments in electrode configurations.”

The remaining comments are minor points (some very minor), primarily concerning the clarity of the manuscript:

We appreciate the reviewer’s suggestions to several detailed corrections that make our manuscript clearer to readers.

1) “Data pre-processing and ROI analysis procedures of spinal data were standard ...”
(how can the authors call this method “standard” if it has only been used in two previous studies?)

The label of our spinal fMRI pre-processing procedure as being “standard” was in reference to commonly used pre-processing steps, such as motion correction, co-registration, and nuisance signal regression, that are similar to what has been done for the past few decades in the brain. However, we do agree that this set of processing steps was developed to minimize physiological noises and to tease out BOLD signal fluctuations specifically in the spinal cord. Thus, we have removed the label of our pre-processing pipeline being “standard” in the revised manuscript.

2) “fMRI images” - the “l” is redundant

This has now been corrected in two instances reflected on line 107 and line 111 in the revised manuscript.

3) Line 100/101: “For each slice, “nuisance” signals derived from muscle and cerebrospinal fluid voxels using principal component analysis.”
This sentence is unclear. Is there a word missing?

Yes, the word “were” is missing and have now been added to the sentence: “For each slice, “nuisance” signals were derived from muscle and cerebrospinal fluid voxels using principal component analysis.” This change is reflected on line 113 in the revised manuscript.

4) In the section “MRI Data Acquisition and Analysis”, the previous published work that established these pre-processing methods for spinal cord fMRI should be cited. There are only two papers cited in this section.

We have now added an additional sentence in this section that cites previous studies in developing pre-processing methods for spinal fMRI with references to two review papers (Eippert et al., 2016; Stroman, 2005).

5) Line 165 (also line 167): “...coherences were also measured...” This seems like vague/strange wording. Do the authors mean specifically correlation between the measured

signals?

Magnitude-squared coherences, which are a function of the power and cross power spectral densities of two signals, were used here to evaluate functional connectivity between electrophysiological signals. We have now added this brief description of what coherence measures to clarify any confusions in lines 183-184.

6) Lines 191-193 (with references removed for clarity): “This method was selected, as were in our previous spinal studies, because ROIs drawn could be made based on hypothesis derived from stimulus-driven data as well as previous fMRI studies in humans and animals.” This sentence is not clear at all.

We agree that this statement is rather vague for readers. Previous studies have shown all four horns of the spinal cord are responsive to stimulus. Specifically, we have previously shown tactile stimulations evoked fMRI responses that were located in the bilateral dorsal and ventral horns. Thus, this leads us to further investigate to what extent these specific ROIs (dorsal and ventral horns) are interconnected in a resting state. Our hypothesis is that neurons engaged in the same function also fluctuated together at rest (an indication of strong resting state functional connectivity). We have now updated this in lines 215-217 of the revised manuscript to clarify this message.

7) Line 194: should the word “aligned” be replaced with “co-registered”, to make the sentence more clear? (If not, then this sentence is not clear).

Yes, that is correct. We have now replaced the word “aligned” with “co-registered” in line 218.

8) Line 197: “...horn-horn connectivity measurements...” This is jargon and should be written in a more clear form. Similarly, several times in the sentence that follow the authors simply refer to “horns”. This is a somewhat sloppy/jargon way to write.

We have now replaced the term “horn-horn connectivity measurements” with “connectivity measurements between ROIs in the spinal cord (dorsal and ventral horns).” Similarly, we have also replaced the term “horn(s)” that follow with “ROI(s)” and indicating that this ROI is “either dorsal or ventral horn of the spinal cord”. We hope this is now clearer to the readers and have resolved the jargon indicated.

9) Line 278: “We found in this study that within-slice (representing one spinal segment) dorsal-dorsal horn resting-state functional connectivity is significantly greater than values for dorsal horn to reference regions, ...”

This phrase would be more clear written as “We found in this study that within-slice (representing one spinal segment) dorsal-dorsal horn resting-state functional connectivity is significantly greater than connectivity values measured between dorsal horn and reference regions, ...”

We have made this change in the revised manuscript reflected in lines 303-306.

10) Line 280: “Quantitative comparisons of connectivity trends ...” I don’t believe the authors actually quantified the connectivity trends. This sentence is unclear.

We compared how connectivity values varied between the three different ROI pairs (dorsal-dorsal within slice, dorsal-intermediate-GM and dorsal-dorsal across slice) for both modalities

and found a highly correlated trend. Using the mean connectivity values from the three ROI pairs from both modalities (Figures 6B and 6C), we computed a Pearson's correlation value ($r=0.9982$). The Pearson's correlation value is now added to Figure 6's legend and clarified in lines 307-308: "Comparison of how connectivity values varied in the three ROI pairs (dorsal-dorsal within slice, dorsal-intermediate-GM and dorsal-dorsal across slice) between the two modalities revealed that they are highly similar." See Reviewer#1's first comment for more details.

11) The paper cited to support the statement that "imaging the cervical spinal cord poses several challenges" is given as Eippert et al., 2016a. This is a very odd choice. There are many papers demonstrating these challenges, and methods to mitigate them, going back many years. A recent review article has summarized them: Powers et al. 2018 (<https://www.ncbi.nlm.nih.gov/pubmed/30201938>)

We would like to thank the reviewer for pointing us to a recent review article on spinal fMRI. We have now included review articles: (Eippert et al., 2016; Powers et al., 2018; Stroman, 2005; Stroman et al., 2014) in lines 394-395 of the revised manuscript.

References for Reviewer#2

- Bannatyne, B.A., Edgley, S.A., Hammar, I., Jankowska, E., Maxwell, D.J., 2003. Networks of inhibitory and excitatory commissural interneurons mediating crossed reticulospinal actions. *Eur. J. Neurosci.* 18, 2273–2284. <https://doi.org/10.1046/j.1460-9568.2003.02973.x>
- Barry, R.L., Smith, S.A., Dula, A.N., Gore, J.C., 2014. Resting state functional connectivity in the human spinal cord. *Elife* 2014, 1–15. <https://doi.org/10.7554/eLife.02812>
- Barry, R.L., Vannesjo, S.J., By, S., Gore, J.C., Smith, S.A., 2018. Spinal cord MRI at 7T. *Neuroimage* 168, 437–451. <https://doi.org/10.1016/j.neuroimage.2017.07.003>
- Eippert, F., Kong, Y., Jenkinson, M., Tracey, I., Brooks, J.C.W., 2016. Denoising spinal cord fMRI data: Approaches to acquisition and analysis. *Neuroimage*. <https://doi.org/10.1016/j.neuroimage.2016.09.065>
- Eippert, F., Kong, Y., Winkler, A.M., Andersson, J.L., Finsterbusch, J., Büchel, C., Brooks, J.C.W., Tracey, I., 2017. Investigating resting-state functional connectivity in the cervical spinal cord at 3 T. *Neuroimage* 147, 589–601. <https://doi.org/10.1016/j.neuroimage.2016.12.072>
- Jankowska, E., 2008. Spinal interneuronal networks in the cat: Elementary components. *Brain Res. Rev.* <https://doi.org/10.1016/j.brainresrev.2007.06.022>
- Powers, M.J., Ioachim, G., Stroman, W.P., 2018. Ten Key Insights into the Use of Spinal Cord fMRI. *Brain Sci.* . <https://doi.org/10.3390/brainsci8090173>
- Stroman, P.W., 2005. Magnetic Resonance Imaging of Neuronal Function in the Spinal Cord: Spinal fMRI. *Clin. Med. Res.* 3, 146–156. <https://doi.org/10.3121/cmr.3.3.146>
- Stroman, P.W.W., Wheeler-Kingshott, C. a., Bacon, M., Schwab, J.M.M., Bosma, R., Brooks, J., Cadotte, D.W., Carlstedt, T., Ciccarelli, O., Cohen-Adad, J., Curt, A., Evangelou, N., Fehlings, M.G.G., Filippi, M., Kelley, B.J.J., Kollias, S., Mackay, A., Porro, C.A. a., Smith, S., Strittmatter, S.M.M., Summers, P., Tracey, I., Stroman, P.W.W., Schwab, J.M.M., Bacon, M., Bosma, R., Brooks, J., Cadotte, D.W., Carlstedt, T., Ciccarelli, O., Cohen-Adad, J., Curt, A., Fehlings, N.E.M.G., Filippi, M., Kelley, B.J.J., Kollias, S., Mackay, A., Porro, C.A. a., Smith, S., Strittmatter, S.M.M., Summers, P., Thompson, A.J., Tracey, I., Evangelou, N., Fehlings, M.G.G., Filippi, M., Kelley, B.J.J., Kollias, S., Mackay, A., Porro,

C.A. a., Smith, S., Strittmatter, S.M.M., Summers, P., Thompson, A.J., Tracey, I., Wheeler-Kingshott, C. a., Bacon, M., Schwab, J.M.M., Bosma, R., Brooks, J., Cadotte, D.W., Carlstedt, T., Ciccarelli, O., Cohen-Adad, J., Curt, A., Evangelou, N., Fehlings, M.G.G., Filippi, M., Kelley, B.J.J., Kollias, S., Mackay, A., Porro, C.A. a., Smith, S., Strittmatter, S.M.M., Summers, P., Tracey, I., Stroman, P.W.W., Schwab, J.M.M., Bacon, M., Bosma, R., Brooks, J., Cadotte, D.W., Carlstedt, T., Ciccarelli, O., Cohen-Adad, J., Curt, A., Fehlings, N.E.M.G., Filippi, M., Kelley, B.J.J., Kollias, S., Mackay, A., Porro, C.A. a., Smith, S., Strittmatter, S.M.M., Summers, P., Thompson, A.J., Tracey, I., 2014. The current state-of-the-art of spinal cord imaging: Applications. *Neuroimage* 84, 1082–1093.
<https://doi.org/10.1016/j.neuroimage.2013.04.124>

Wu, T.-L., Wang, F., Mishra, A., Wilson, G.H., Byun, N., Chen, L.M., Gore, J.C., 2017. Resting-state functional connectivity in the rat cervical spinal cord at 9.4 T. *Magn. Reson. Med.*
<https://doi.org/10.1002/mrm.26905>

Reviewer #3 (Remarks to the Author):

Wu et al., investigated spinal gray matter functional connectivity via both depth electrophysiological recordings and fMRI in monkeys to validate rsfMRI for assessing spinal FC and to claim basic intrinsic functional architecture in spinal neuronal activity in resting state. By comparison of rsfMRI to LFP and MUA signals across dorsal-intermediate-ventral and adjacent segments along spinal cord based on ROI method, a well agreement observed in spatial connectivity pattern. The comparison conducted on both sensory stimulation and analysis of spontaneous activity in these regions. The comparison experimental design is strict. Relative to brain fMRI, spinal segment may be much smaller and challenge to have high spatial information on tiny gray matter. For example, along the gray matter of dorsal to ventral in imaging plane, only few EPI voxels, so partial volume effects may be a potential issue for precision of mapping functional activity, but ROI may be better alternative to choose as authors did. However, ROI analysis may lose detailed spatial information on functional architecture.

line 89/90, TR=46.9ms, ~3s/volume, some details need to be clear on the fMRI scan.

We have now added more details about the fMRI scan on lines 93-94 in the revised manuscript: “fast gradient echo sequence (TR/TE=46.9/6.50ms, matrix size=64 x 64, field of view=32 x 32 mm², resolution = 0.5 x 0.5 x 3 mm³, flip angle~15°, ~3s/volume).

line 112, LFP and MUA recording parameters need details.

LFP signals were recorded continuously with a sampling rate of 500Hz and low pass filtered at 250Hz. MUA was also recorded by capturing time stamps of spikes evoked using the same system. Spike processing employed a bandpass filter of 250-5000Hz while a global spike detection threshold was set to -4 times the root mean squared energy. This additional information has now been added to the revised manuscript in lines 139-143.

figure 4 showing % neural signal changes, how did you calculate the percentage for LFP and MUA?

For MUA, percentage neural signal changes were calculated by finding the percentage change in spike rate at each timepoint during stimulus-on relative to the averaged spike rate during pre-stimulus baseline signals. Pre-stimulus baseline signals were defined as the spike rate during baseline period of 2 seconds preceding stimulation. This can be expressed mathematically as:

$$\frac{Spike\ Rate_{stimulus-on} - Spike\ Rate_{pre-stimulus}}{Spike\ Rate_{stimulus-off}}$$

Similarly, for LFP, percentage changes were calculated by finding the percentage change in group LFP signals at each timepoint during stimulus-on relative to averaged LFP signals during pre-stimulus baseline signals. Pre-stimulus baseline signals were defined as the LFP signal amplitude during baseline period of 3 seconds preceding stimulation. This can also be expressed mathematically as:

$$\frac{LFP_{stimulus-on} - LFP_{pre-stimulus}}{LFP_{stimulus-off}}$$

We realize the same baseline time period for both LFP and MUA should have been extracted, so we have now updated the spike rate baseline period to 3 seconds (instead of 2 seconds) preceding stimulation for performing the calculation. Additionally, we also noticed a minor bug in our code where not all epochs were captured in the group averaged calculation. These have now been fixed and updated and corrected in Figure 4 and on line in the revised manuscript with source data uploaded:

Figure 4: Responses to innocuous tactile stimulation of digits in the spinal cord.

line 245, the conclusion of U-shaped pattern can not be exactly drew without clear functional mapping, or just based on few ROI analysis.

We agree that the U-shaped pattern observed warrants further investigation given that this was observed in only over half the runs (58% of the runs), and especially this phenomenon has yet to be reported. However, group functional mapping with seed activation map in the dorsal horn presents patterns consistent with such observation, as displayed in our previous publication (Figure 2 in Chen et al., 2015). That being said, functional mapping with electrophysiology recordings would provide even more convincing evidence which, unfortunately, was not made possible given our limited field of view without sampling the ventral horns (see Reviewer#2's first comment). Overall, while we agree that further investigations can be made to more accurately depict how functional connectivity varies as a function of depth in the spinal cord (which may require imaging at a higher spatial resolution, see reviewer's last comment), findings here present a first-hand analysis in observing such trends.

The fMRI and ephys were not simultaneously obtained, some details needed on how well these experiments were controlled in same physiological conditions and anesthesia states.

Yes, indeed, maintaining the same physiological conditions and anesthesia states is critical when making comparisons of functional connectivity between the two modalities. While the exact levels of isoflurane varied slightly across experiments (0.7-0.8%), for both MRI and electrophysiology, animals were maintained in almost identical physiological states with vital signal readings of peripheral oxygen saturation rate at >98%, heart rate (260-280 bpm), end-tidal CO₂ (35-45 mmHg) and rectal temperature (37.5-38.5°C). This was perhaps not emphasized in our initial submission and we have now added a clarification sentence that the anesthesia and physiological conditions were kept comparable when acquiring data for the two modalities.

The ephys recording and fMRI scans were conducted under isoflurane, certainly anesthesia effects may present for functional connectivity. Related discussion, especially refer to simultaneous LFP/fMRI animal brain studies, will help.

Yes, this is a good point brought up by the reviewer. We have now discussed the potential influence of anesthesia on functional connectivity in the last paragraph of “Challenges of imaging and electrophysiological recordings in the spinal cord and limitations” under the Discussion section in lines 424-438.

“Finally, a possible confounding factor is the influence of anesthesia in the measurement of functional connectivity for both modalities. An increase in anesthesia level has been shown to reduce cortical functional connectivity in non-human primates (Hutchison et al., 2014; Wu et al., 2016) while others have indicated the possibility of neurovascular decoupling at high anesthesia dosages (see a review Masamoto and Kanno, 2012). That being said, studies have also demonstrated that anesthesia has a weaker influence in early sensory cortical regions (Bonhomme et al., 2011) and thus, this effect is most likely reduced in the downstream spinal cord. In our experiment, the isoflurane was maintained at less than 1%, which is lower than the range where significant neuronal connectivity activity drops and becomes unstable in the brain (1.75% isoflurane reported in (Hutchison et al., 2014)). In fact, a recent study found that neurovascular coupling in the spinal cord remains unaltered at 1.2% isoflurane in an experiment with decerebrated rats undergoing electrical stimulation. Moreover, simultaneous recordings of local evoked field potentials and fMRI in brain studies have shown signals from the two modalities are still coupled under anesthesia (Brinker et al., 1999; Gsell et al., 2006; Huttunen et al., 2008; N. K. Logothetis et al., 2001; Shmuel et al., 2006). With this information, we believe anesthesia is unlikely a major confounding factor to the trends observed in this study.”

The limitations and challenges may include fMRI spatial resolution due to tiny spinal gray matter. Potential improvement in discussion may be appreciated.

Agreed. We have now added the following discussion as an additional paragraph in “Challenges of imaging and electrophysiological recordings in the spinal cord and limitations” under the Discussion section in lines 416-421.

“Achieving high spatial resolution with adequate SNR in fMRI is critical for minimizing partial volume effects as well as for characterizing functional connectivity at fine scale. In this study, high SNR and resolution were achieved by imaging a stable preparation at 9.4T, by acquiring images with a multi-shot gradient-echo sequence (a technique used in Barry et al., 2016, 2014) and by reducing TE (a technique used in Zhao et al., 2009, 2008). Further potential improvements may be obtained using better RF coils (e.g. cryogenic coils) and deploying more advanced shimming methods.”

References for Reviewer#3

- Barry, R.L., Rogers, B.P., Smith, S.A., Gore, J.C., 2016. Reproducibility of resting state spinal cord networks at 7 Tesla. *Neuroimage* 133, 31–40. <https://doi.org/10.1016/j.neuroimage.2016.02.058>
- Barry, R.L., Smith, S.A., Dula, A.N., Gore, J.C., 2014. Resting state functional connectivity in the human spinal cord. *Elife* 2014, 1–15. <https://doi.org/10.7554/eLife.02812>
- Bonhomme, V., Boveroux, P., Hans, P., Brichant, J.F., Vanhaudenhuyse, A., Boly, M., Laureys, J.

- S., 2011. Influence of anesthesia on cerebral blood flow, cerebral metabolic rate, and brain functional connectivity. *Curr. Opin. Anaesthesiol.* <https://doi.org/10.1097/ACO.0b013e32834a12a1>
- Brinker, G., Bock, C., Busch, E., Krep, H., Hossmann, K.A., Hoehn-Berlage, M., 1999. Simultaneous recording of evoked potentials and T2*-weighted MR images during somatosensory stimulation of rat. *Magn. Reson. Med.* [https://doi.org/10.1002/\(SICI\)1522-2594\(199903\)41:3<469::AID-MRM7>3.0.CO;2-9](https://doi.org/10.1002/(SICI)1522-2594(199903)41:3<469::AID-MRM7>3.0.CO;2-9)
- Chen, L.M., Mishra, A., Yang, P.-F., Wang, F., Gore, J.C., 2015. Injury alters intrinsic functional connectivity within the primate spinal cord. *Proc. Natl. Acad. Sci. U. S. A.* 112, 5991–6. <https://doi.org/10.1073/pnas.1424106112>
- Gsell, W., Burke, M., Wiedermann, D., Bonvento, G., Silva, A.C., Dauphin, F., Buhle, C., Hoehn, M., Schwindt, W., 2006. Differential Effects of NMDA and AMPA Glutamate Receptors on Functional Magnetic Resonance Imaging Signals and Evoked Neuronal Activity during Forepaw Stimulation of the Rat. *J. Neurosci.* <https://doi.org/10.1523/JNEUROSCI.4615-05.2006>
- Hutchison, R.M., Hutchison, M., Manning, K.Y., Menon, R.S., Everling, S., 2014. Isoflurane induces dose-dependent alterations in the cortical connectivity profiles and dynamic properties of the brain's functional architecture. *Hum. Brain Mapp.* 35, 5754–5775. <https://doi.org/10.1002/hbm.22583>
- Huttunen, J.K., Gröhn, O., Penttonen, M., 2008. Coupling between simultaneously recorded BOLD response and neuronal activity in the rat somatosensory cortex. *Neuroimage* 39, 775–785. <https://doi.org/10.1016/j.neuroimage.2007.06.042>
- Logothetis, N.K., Pauls, J., Augath, M., Trinath, T., Oeltermann, A., 2001. Neurophysiological investigation of the basis of the fMRI signal. *Nature* 412, 150–157. <https://doi.org/10.1038/35084005>
- Masamoto, K., Kanno, I., 2012. Anesthesia and the quantitative evaluation of neurovascular coupling. *J. Cereb. Blood Flow Metab.* <https://doi.org/10.1038/jcbfm.2012.50>
- Shmuel, A., Augath, M., Oeltermann, A., Logothetis, N.K., 2006. Negative functional MRI response correlates with decreases in neuronal activity in monkey visual area V1. *Nat Neurosci* 9, 569–577. <https://doi.org/10.1038/nn1675>
- Wu, T.-L., Mishra, A., Wang, F., Yang, P.-F., Gore, J.C., Chen, L.M., 2016. Effects of isoflurane anesthesia on resting-state fMRI signals and functional connectivity within primary somatosensory cortex of monkeys. *Brain Behav.* 6, e00591. <https://doi.org/10.1002/brb3.591>
- Zhao, F., Williams, M., Meng, X., Welsh, D.C., Coimbra, A., Crown, E.D., Cook, J.J., Urban, M.O., Hargreaves, R., Williams, D.S., 2008. BOLD and blood volume-weighted fMRI of rat lumbar spinal cord during non-noxious and noxious electrical hindpaw stimulation. *Neuroimage* 40, 133–147. <https://doi.org/10.1016/j.neuroimage.2007.11.010>
- Zhao, F., Williams, M., Meng, X., Welsh, D.C., Grachev, I.D., Hargreaves, R., Williams, D.S., 2009. Pain fMRI in rat cervical spinal cord: An echo planar imaging evaluation of sensitivity of BOLD and blood volume-weighted fMRI. *Neuroimage* 44, 349–362. <https://doi.org/10.1016/j.neuroimage.2008.09.001>

REVIEWERS' COMMENTS:

Reviewer #1 (Remarks to the Author):

The authors should be congratulated for performing such a comprehensive set of experiments.

I have only one comment: in the rebuttal you mention that you have now used FDR to adjust significance values. However, I would be interested to know the effect of a more conventional FWE correction such as Bonferroni. I.e. what are the total number of comparisons performed outside of the dorsal seed region? Related to this, currently it's not clear to me what is meant by "we performed a FDR correction for the number of statistical comparisons made with the group of connectivity values defined at the intermediate GM". Does this mean that you adjust according to the number of voxels (i.e. connectivity values) at a single location? What was the rationale behind that choice?

Jonathan Brooks

Reviewer #2 (Remarks to the Author):

I feel that the authors have adequately responded to all of the points I raised in my initial review.

Reviewer #3 (Remarks to the Author):

The ephys recordings were sampled in 500Hz, that would be no problem for <250Hz LFP analysis, but problem for >250Hz analysis of MUA, 250-5000Hz as claimed in paper. Pleas double check.

RESPONSES TO REVIEWERS

Reviewer #1 (Remarks to the Author):

The authors should be congratulated for performing such a comprehensive set of experiments.

I have only one comment: in the rebuttal you mention that you have now used FDR to adjust significance values. However, I would be interested to know the effect of a more conventional FWE correction such as Bonferroni. I.e. what are the total number of comparisons performed outside of the dorsal seed region? Related to this, currently it's not clear to me what is meant by "we performed a FDR correction for the number of statistical comparisons made with the group of connectivity values defined at the intermediate GM". Does this mean that you adjust according to the number of voxels (i.e. connectivity values) at a single location? What was the rationale behind that choice?

Jonathan Brooks

We agree with the reviewer that this should be clarified in our manuscript. The p-values indicated on Figure 5C were derived from two-sided Mann-Whitney Tests when comparing groups of r-values at a particular distance away from the intermediate (I-GM) relative to the group of r values at I-GM (0 voxels away). A total of eight comparisons were performed (-5 voxels away to 3 voxels away, excluding 0 voxel away) and thus we performed a FDR correction based on these eight comparisons. While we understand this is not a correction in the strictest sense (i.e. the correction was made to the number of ROI comparisons after computation of r-values), we believe this information is relevant as it conveys how different correlation values are to each other at different voxel depths relative to the I-GM. We have now also added a Bonferroni correction to the p-values from the two-sided Mann-Whitney Test below as well:

Table#R1: P-values of two-sided Mann-Whitney statistical comparison tests between groups of connectivity values at various depths relative to the I-GM from Figure 5C

	Voxels away from intermediate GM								
	-5	-4	-3	-2	-1	0	1	2	3
p-value (uncorrected)	1.67×10^{-10}	1.27×10^{-11}	4.62×10^{-11}	2.68×10^{-7}	2.70×10^{-2}	N/A	7.55×10^{-1}	8.14×10^{-2}	2.53×10^{-2}
p-values (FDR-corrected)	5.02×10^{-10}	1.14×10^{-10}	2.08×10^{-10}	6.02×10^{-7}	4.05×10^{-2}	N/A	7.55×10^{-1}	9.15×10^{-2}	4.05×10^{-2}
p-values (Bonferroni corrected)	1.34×10^{-9}	1.02×10^{-10}	3.69×10^{-10}	2.14×10^{-6}	2.16×10^{-1}	N/A	6.04×10^0	6.51×10^{-1}	2.02×10^{-1}

We can also assess whether the correlation r-values are significant and perform corrections at the voxel level, which seems to be what the reviewer may be suggesting here, and we did so additionally in this revision. We first evaluated the p-values derived from the r-values, at different voxel depths, with the dorsal horn as the seed. Given muscle and CSF signals were regressed from the time series in pre-processing, we subsequently performed Bonferroni corrections

based on the number of voxels in gray matter outside the dorsal seed region for each animal (averaged number of voxels for correction \pm standard deviation = 85.36 ± 12.25 voxels):

Table#R2: P-values of Pearson's r-values corrected and uncorrected from Figure 5C

	Voxels away from intermediate GM								
	-5	-4	-3	-2	-1	0	1	2	3
p-value (uncorrected)	0	1.96×10^{-7}	1.51×10^{-5}	2.05×10^{-4}	3.59×10^{-4}	1.43×10^{-2}	1.61×10^{-2}	9.07×10^{-3}	2.82×10^{-5}
p-values (Bonferroni corrected)	0	1.70×10^{-5}	1.31×10^{-3}	1.78×10^{-2}	3.12×10^{-2}	1.15×10^0	1.39×10^0	7.36×10^{-1}	2.45×10^{-3}

We have now clarified what that the p-values in the figure legend of Figure 5C represent, and included both tables above in the Supplementary Information.

Reviewer #2 (Remarks to the Author):

I feel that the authors have adequately responded to all of the points I raised in my initial review.

Reviewer #3 (Remarks to the Author):

The ephys recordings were sampled in 500Hz, that would be no problem for <250Hz LFP analysis, but problem for >250Hz analysis of MUA, 250-5000Hz as claimed in paper. Pleas double check.

We apologize for the confusion caused here. MUA processing was performed by the system using digitized signals sampled at 30kHz that were input from the digital headstage, which allowed us to record time stamps of spikes evoked. Concurrently, the system also records the unfiltered continuous signals sampled at 30kHz as well as the filtered continuous signal with a lower sampling rate. The latter was what we recorded for LFP analysis with a sampling rate of 500Hz selected. We have now added this information to the revised manuscript in lines 483-484 of the clean manuscript.